# STLA: Spatiotemporal Lookahead Alignment for Post-Training Quantization

Zuqi Zhang [1]   Chenghe Sun [1]   Xiangyi Chu [1]   Wei-Han Yu [1]   Ka-Fai Un [1]   Rui P. Martins [1]   Pui-In Mak [1]   Jiawei Xu [1]

## Abstract

Adaptive rounding techniques in Post-Training Quantization (PTQ) enable the efficient deployment of Large Language Models (LLMs) with low resource and data dependencies. While learning-based rounding methods are accurate yet costly, compensation-based approaches offer a highly efficient alternative. However, synergizing these two to realize their full potential is hindered by spatiotemporal misalignment in the decoupled paradigm. Key challenges include temporal parameter conflict, the invalidation of the initial Round-to-Nearest (RTN) assumption, and spatially-inconsistent optimization objectives. This paper introduces STLA, a novel rounding-optimized PTQ framework that achieves both fast and accurate LLM quantization. STLA resolves temporal inconsistency through cluster-wise integrated rounding optimization, which collocates the learning and compensation phases. STLA achieves spatial alignment through a unified global objective derived from the Schur Complement, enabling the solver to look ahead and align local rounding decisions with the optimal future compensation of remaining weights. Furthermore, we propose a Hessian-guided clustering strategy that exploits both diagonal and off-diagonal information to maximize intra-cluster error cancellation. Extensive experiments demonstrate that STLA establishes a new state-of-the-art for low-bit PTQ while maintaining high computational efficiency. The code is available at https://github.com/AI2C-Lab/STLA.

[1] State-Key Laboratory of Analog and Mixed-Signal VLSI/Institute of Microelectronics and Faculty of Science and Technology, University of Macau, Macau, China. Correspondence to: Jiawei Xu <jiaweixu@um.edu.mo>.

*Proceedings of the 43rd International Conference on Machine Learning*, Seoul, South Korea. PMLR 306, 2026. Copyright 2026 by the author(s).

## 1. Introduction

The scaling of Large Language Models (LLMs), such as the LLaMA (Touvron et al., 2023) and OPT series (Zhang et al., 2022), has revolutionized natural language processing but also introduced massive computational and memory demands that strain existing hardware infrastructure. With hundreds of billions of parameters, effective model compression becomes essential, especially for deployment on edge devices where bandwidth and memory are key bottlenecks.

Quantization compresses deep neural networks (DNNs) by reducing the precision of weights and activations. While Quantization-Aware Training (QAT) (Liu et al., 2024) achieves high accuracy, its reliance on full-dataset retraining makes it computationally expensive, time-consuming, and privacy-invasive, particularly for LLMs. Consequently, Post-Training Quantization (PTQ) (Nagel et al., 2020; Li et al., 2021; Frantar et al., 2023; Li et al., 2025; Xu et al., 2026) has emerged as a promising solution, enabling fast and efficient compression with minimal calibration data. Representative approaches include AdaRound (Nagel et al., 2020), which learns optimal rounding decisions beyond rounding-to-nearest (RTN), and GPTQ (Frantar et al., 2023), which exploits second-order Hessian information for sequential error compensation.

Theoretically, combining the speed of error compensation with the precision of learning-based rounding would yield optimal performance. Yet, existing PTQ strategies fall short, as they either utilize these techniques in isolation or adopt a decoupled "compensate-then-learn" paradigm. We identify that this decoupled nature leads to intrinsic conflicts. **Temporal Misalignment**: Compensation updates invalidate the initial optimal quantization parameters, and the compensation itself becomes invalid once the learning phase overturns the critical RTN assumption. **Spatial Misalignment**: Local rounding decisions often neglect the global information of the loss landscape, causing the optimization to settle for local optima rather than the global best, thereby degrading accuracy. Resolving these spatiotemporal misalignments to achieve fast and accurate PTQ remains an open challenge.

In this paper, we propose **S**patio**T**emporal **L**ookahead **A**lignment (**STLA**), a novel PTQ framework that overcomes spatiotemporal misalignments. Our main contributions are:

- We resolve temporal misalignment by collocating the learning and compensation phases. This is realized through a novel strategy that couples intra-cluster rounding learning with cluster-based coarse-grained compensation, efficiently enabled by Cholesky-factored triangular updates.

- We propose a spatial alignment strategy that optimizes a lookahead global objective derived from the Schur complement of the inverse Hessian. We enhance this with a novel Hessian-guided clustering strategy that utilizes diagonal sensitivity and off-diagonal correlations to maximize intra-cluster error cancellation.

- We integrate these components into a unified PTQ framework to achieve fast and accurate LLM PTQ, bridging the gap between the efficiency of compensation-based solvers and the accuracy of learning-based rounding.

## 2. Related Work

**PTQ.** QAT (Liu et al., 2024; Chen et al., 2025) embeds quantization into training and achieves strong accuracy, but its retraining cost and reliance on labeled data limit its applicability to large neural network models. In contrast, PTQ requires only a small unlabeled calibration set, enabling fast and efficient low-precision weight and/or activation deployment (Nagel et al., 2020; Li et al., 2021). Weight quantization compresses static parameters before deployment to alleviate memory bottlenecks, whereas activation quantization targets faster computation at runtime during inference. While activation PTQ can benefit from methods such as LSQ (Esser et al., 2020) and SmoothQuant (Xiao et al., 2023), weight PTQ can further exploit advanced learning-based (Nagel et al., 2020; Li et al., 2021; Lee et al., 2023) and compensation-based (Frantar & Alistarh, 2022; Frantar et al., 2023; Li et al., 2025) element-wise rounding strategies that are inherently incompatible with dynamic activations. Building on learning-based and compensation-based PTQ methods, this work specifically focuses on weight quantization.

Calibration in recent PTQ advances shifts the optimization objective from naive weight error $\|\mathbf{W} - \widehat{\mathbf{W}}\|_F^2$ to the task-aware reconstruction error $\|\mathbf{W}\mathbf{X} - \widehat{\mathbf{W}}\mathbf{X}\|_F^2$, leveraging calibration data to capture input-dependent sensitivity to weight rounding noise. Going beyond RTN, AdaRound (Nagel et al., 2020) formulates quantization as a differentiable optimization of element-wise rounding decisions. Building on this, BRECQ (Li et al., 2021) extends the reconstruction error to residual blocks to capture inter-layer dependencies, and FlexRound (Lee et al., 2023) enlarges the learning space by reparametrizing rounding decisions using element-wise division. These learning-based methods exhibit robustness

across architectures, such as BRECQ for Convolutional Neural Networks (CNNs) and APHQ-ViT (Wu et al., 2025) for Vision Transformers.

**PTQ for LLMs.** The explosive growth in model parameters and computational cost of emerging LLMs poses new challenges for PTQ, highlighting the need for speed and efficiency. To address this, various LLM-oriented PTQ methods have recently been proposed. VS-Quant (Dai et al., 2021) first introduced fine-grained group-wise quantization by partitioning weights into smaller vector segments. Z-Fold (Jeon et al., 2023) leverages the Transformer architecture to optimize parameter folding strategies. GPTQ (Frantar et al., 2023) achieves fast and accurate PTQ by utilizing approximate second-order Hessian information for iterative weight compensation, a paradigm recently advanced by GPTAQ (Li et al., 2025) via asymmetric calibration. To mitigate outliers, QuIP (Chee et al., 2023) combines Hessian-guided compensation with randomized orthogonal rotations. AWQ (Lin et al., 2024) emphasizes the critical role of activations in weight PTQ and protects activation-derived salient weights through scaling-up.

Learning-based PTQ methods (Nagel et al., 2020; Li et al., 2021) achieve high accuracy, but their memory footprint and computational overhead become bottlenecks when scaled to LLMs. For example, BRECQ requires 27.8 GB of GPU memory and 19.2 hours of runtime to quantize a 2.7B OPT model (Kim et al., 2024), making it prohibitively expensive compared to compensation-based methods (Frantar et al., 2023; Li et al., 2025). To reduce this complexity, AESPA (Kim et al., 2024) adopts a divide-and-conquer strategy, decomposing hyper-scale Transformer reconstruction into sub-problems to improve efficiency. However, efforts are still needed to address the fundamental challenges of temporal and spatial misalignment.

## 3. Background

**Notations.** Scalars, vectors, and matrices are denoted by non-bold lowercase (e.g., $s$), bold lowercase (e.g., $\mathbf{w}$), and bold uppercase letters (e.g., $\mathbf{W}$), respectively. We adopt row-vector notation where $\mathbf{w}$ represents a single column of the weight matrix $\mathbf{W}$. $\widehat{\mathbf{W}}$ denotes the quantized $\mathbf{W}$. $\mathbf{X}$ denotes the input activation matrix, while $\widetilde{\mathbf{X}}$ denotes the floating-point model's layer inputs. The Hessian matrix and its inverse are denoted as $\mathbf{H}$ and $\mathbf{H}^{-1}$. In this paper, $\lfloor\cdot\rceil$ denotes the RTN operator, $\lfloor\cdot\rfloor$ denotes the floor operator, and $\|\cdot\|_F$ signifies the Frobenius norm.

**Sequential Compensation-based Rounding.** OBQ (Frantar & Alistarh, 2022) employs an iterative greedy strategy. In each iteration, it identifies the quantized weight $w_q$ that minimizes the quantization error and calculates the optimal compensation update $\boldsymbol{\delta}$ for the remaining unquantized

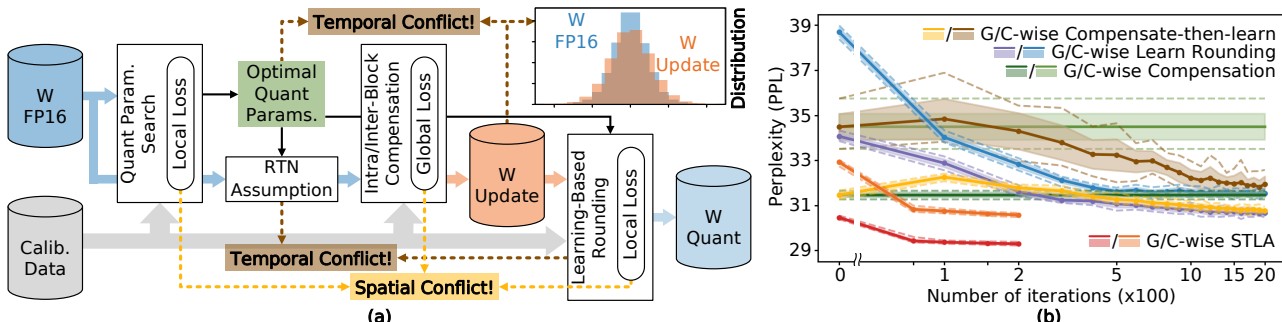

*Figure 1.* Illustration of the spatiotemporal misalignments in the decoupled paradigm. (a) Conflicts arise from spatial misalignment (local vs. global objectives) and temporal misalignment (invalidation of optimal quantization parameters and the RTN assumption). (b) OPT-125M channel (C)-wise / group (G)-wise C4 Perplexity (PPL) curves confirm that the decoupled "compensate-then-learn" paradigm (brown / yellow) struggles, failing to outperform standalone learning-based rounding (blue / purple). STLA (orange / red) resolves these conflicts, enabling faster convergence and reduced PPL.

weights:

$$w_q = \arg\min_{w_q} \frac{(\text{quant}(w_q) - w_q)^2}{(\mathbf{H}^{-1})_{qq}},$$

$$\boldsymbol{\delta} = -\frac{w_q - \text{quant}(w_q)}{(\mathbf{H}^{-1})_{qq}} \cdot (\mathbf{H}^{-1})_{:,q}. \tag{1}$$

Following this, the inverse Hessian is updated via Gaussian elimination. GPTQ (Frantar et al., 2023) scales this to LLMs using block-wise updates and Cholesky decomposition. GP-TAQ (Li et al., 2025) further incorporates input residuals $\mathbf{r} = \mathbf{w}\widetilde{\mathbf{X}} - \mathbf{w}\mathbf{X}$ via asymmetric calibration:

$$\boldsymbol{\delta}_{\text{GPTAQ}} = -\frac{w_q - \text{quant}(w_q)}{(\mathbf{H}^{-1})_{qq}} \cdot (\mathbf{H}^{-1})_{:,q} + \mathbf{r}\mathbf{X}^\top \mathbf{H}^{-1}. \tag{2}$$

**Learning-based Adaptive Rounding.** AdaRound (Nagel et al., 2020) demonstrates that RTN is suboptimal and proposes learning the rounding decision. It optimizes a continuous variable $\mathbf{v}$ by minimizing the reconstruction objective:

$$\arg\min_{\mathbf{v}} \Delta\mathbf{w} \cdot \mathbf{H} \cdot \Delta\mathbf{w}^\top + \lambda f_{reg}(\mathbf{v}), \tag{3}$$

where the quantization noise is defined as $\Delta\mathbf{w} = \mathbf{w} - (\lfloor\mathbf{w}\rfloor + h(\mathbf{v}))$. Here, $h(\mathbf{v})$ is a rectified sigmoid function for differentiable optimization, and $f_{reg}(\mathbf{v})$ is a regularization term that encourages the continuous variables to converge towards binary values (0 or 1) to ensure valid final rounding.

**The Decoupled Paradigm.** Ideally, integrating the efficiency of compensation mechanisms with the accuracy benefits of learning-based rounding offers a synergistic solution. However, the existing two-stage baseline (Kim et al., 2024) fails to realize this potential. In its decoupled pipeline, GPTQ first performs sequential error compensation for each element under an RTN assumption, after which AdaRound

learns element-wise optimal rounding decisions in a layer-wise manner. As shown in Figure 1, the "compensate-then-learn" paradigm suffers from fundamental spatiotemporal misalignments.

## 4. Methodology

We identify spatial and temporal misalignments as the critical bottlenecks in existing decoupled paradigms. To address these challenges, we propose the STLA framework, which reformulates the PTQ workflow through a spatiotemporal optimization as illustrated in Figure 2. STLA further extends Hessian-guided clustering to enhance optimization performance. We also provide a comprehensive complexity analysis, demonstrating that STLA achieves these performance gains with minimal computational overhead.

### 4.1. Motivation

We identify that existing decoupled paradigms suffer from intrinsic conflicts along two dimensions: temporal misalignment and spatial misalignment.

**Temporal Misalignment.** The first temporal conflict emerges when the compensated weight $\widehat{w}_{rtn}$ relies on an RTN assumption that is later overturned by the decoupled adaptive learning phase ($\widehat{w}_{ada}$). This reversal invalidates the precondition of the prior compensation, injecting an irreducible drift vector $\boldsymbol{\Delta}$ into the weight compensation:

$$\boldsymbol{\Delta} = \frac{\widehat{w}_{rtn} - \widehat{w}_{ada}}{(\mathbf{H}^{-1})_{qq}} \cdot (\mathbf{H}^{-1})_{q:,q}. \tag{4}$$

This implies that weights are adjusted to compensate for an inaccurately estimated residual. Given that the true residual is only revealed after the learning-based rounding phase, this conflict remains fundamentally irreducible within the decoupled "compensate-then-learn" paradigm.

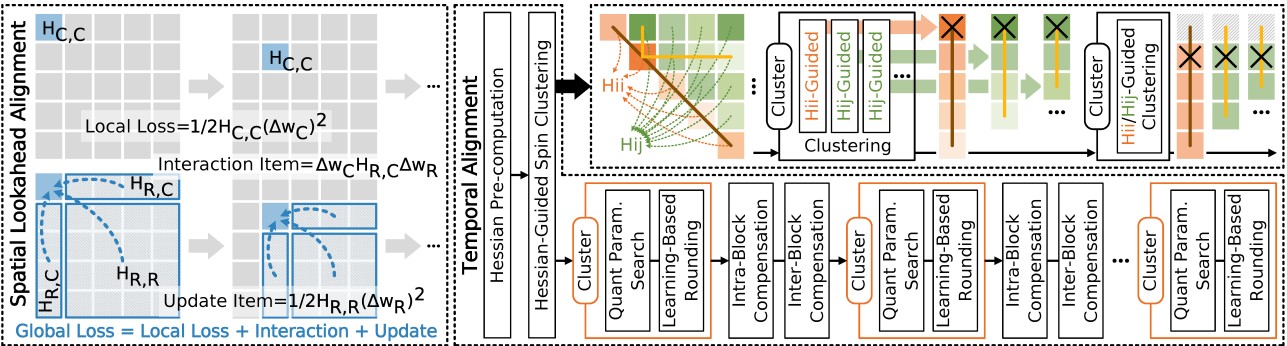

*Figure 2.* The overall STLA framework consists of two major strategies targeting spatial and temporal conflicts. Spatial alignment unifies a lookahead global objective across all optimization components. Temporal alignment restructures the PTQ pipeline to synergistically collocate learning and compensation in a cluster-wise manner. To enhance intra-cluster error cancellation of this coupled paradigm, a Hessian-guided clustering component is integrated into the pre-computation stage.

Another temporal conflict arises as continuous error propagation via off-diagonal couplings induces non-stationarity in the weight distribution. Consequently, the pre-computed static quantization parameters become misaligned with the evolving compensated weights.

**Spatial Misalignment.** In the PTQ frameworks shown in Figure 1, components such as quantization parameter search, compensation and learning-based rounding all involve optimization processes targeting specific objectives. For each group or cluster, existing approaches like (Lin et al., 2024) typically minimize a local reconstruction error governed by the Hessian submatrix $\mathbf{H}_{\mathcal{C},\mathcal{C}}$ of the current cluster:

$$\mathcal{L}_{local} = \Delta \mathbf{w}_{\mathcal{C}} \cdot \mathbf{H}_{\mathcal{C},\mathcal{C}} \cdot \Delta \mathbf{w}_{\mathcal{C}}^{\top}. \quad (5)$$

This objective is theoretically suboptimal as it optimizes the subspace in isolation, implicitly assuming that surrounding weights remain frozen. This formulation leads to spatial misalignment by neglecting the critical correlations with the remaining weights.

### 4.2. STLA Framework

Addressing the misalignments detailed in 4.1, STLA reformulates quantization as a unified optimization problem over high-dimensional subspaces (Figure 2). It integrates parameter search, learning-based rounding, and compensation within a unified cluster-wise framework. Specifically, via the proposed spin clustering strategy, the entire weight matrix is partitioned into clusters based on Hessian-guided correlations. For each cluster, we apply a quantization parameter search, and then a learning-based rounding process guided by a lookahead global objective, reflecting the curvature of the loss landscape. Once the optimal discrete configuration for a cluster is determined, the resulting quantization error is propagated to the remaining weights in a single update. This mechanism ensures that local rounding

decisions remain strictly aligned with the global reconstruction goal.

We define the cluster ($\mathcal{C}$) as the fundamental subspace within which joint optimization is performed. It is crucial to distinguish this term from parameter-sharing "group" or hardware-buffering "block" (see Appendix B for a detailed distinction). Formally, at each optimization step, we partition the remaining unquantized weights into the current cluster $\mathcal{C}$ (comprising $c$ weights) and the remaining set $\mathcal{R}$.

### 4.3. Lookahead Global Objective

Inspired by OBQ, we derive a lookahead global objective via the Schur complement. This approach projects the global loss landscape onto the local cluster subspace by anticipating the optimal response of the remaining weights.

Let $\mathcal{C}$ and $\mathcal{R}$ denote the column indices of the current cluster and the remaining weights, respectively. Standard solvers minimize a local proxy loss $\mathcal{L}_{local} = \Delta \mathbf{w}_{\mathcal{C}} \mathbf{H}_{\mathcal{C}} \Delta \mathbf{w}_{\mathcal{C}}^{\top}$, which suffers from the spatial misalignment discussed in 4.1. To align the current decision with the global loss landscape, we derive the lookahead global objective shown in Figure 2. By anticipating the optimal future compensation of $\mathcal{R}$, the global error introduced by quantizing $\mathcal{C}$ collapses to:

$$\mathcal{L}_{global}(\Delta \mathbf{w}_{\mathcal{C}}) = \Delta \mathbf{w}_{\mathcal{C}} \left( (\mathbf{H}^{-1})_{\mathcal{C},\mathcal{C}} \right)^{-1} \Delta \mathbf{w}_{\mathcal{C}}^{\top}, \quad (6)$$

where $\Delta \mathbf{w}_{\mathcal{C}}$ is the quantization error, and $(\mathbf{H}^{-1})_{\mathcal{C},\mathcal{C}}$ denotes the submatrix of the inverse Hessian corresponding to the cluster indices. This objective mathematically embodies the current global error, strictly generalizing the scalar metric from OBQ to the subspace level. See Appendix A.1 for the detailed derivation.

**Numerical Stability via Cholesky Decomposition.** Direct evaluation of Eq. (6) requires a double inversion of the Hessian submatrix, a process prone to numerical instability

and floating-point error accumulation. To prevent this, we utilize the Cholesky decomposition of the inverse Hessian, denoted as $\mathbf{H}^{-1} = \mathbf{L}\mathbf{L}^\top$. Let $\mathbf{L}_\mathcal{C}$ be the Cholesky factor block corresponding to the current cluster. The objective can be reformulated as:

$$
\begin{aligned}
\mathcal{L}_{global}(\Delta\mathbf{w}_\mathcal{C}) &= \Delta\mathbf{w}_\mathcal{C} \left(\mathbf{L}_\mathcal{C}\mathbf{L}_\mathcal{C}^\top\right)^{-1} \Delta\mathbf{w}_\mathcal{C}^\top, \\
&= \Delta\mathbf{w}_\mathcal{C}(\mathbf{L}_\mathcal{C}^\top)^{-1}\mathbf{L}_\mathcal{C}^{-1}\Delta\mathbf{w}_\mathcal{C}^\top, \\
&= \left(\mathbf{L}_\mathcal{C}^{-1}\Delta\mathbf{w}_\mathcal{C}^\top\right)^\top \left(\mathbf{L}_\mathcal{C}^{-1}\Delta\mathbf{w}_\mathcal{C}^\top\right), \\
&= \left\|\mathbf{L}_\mathcal{C}^{-1}\Delta\mathbf{w}_\mathcal{C}^\top\right\|_F^2.
\end{aligned} \tag{7}
$$

In practice, we avoid explicit matrix inversion of $\mathbf{L}_\mathcal{C}$ entirely. Instead, we compute the term $\mathbf{y} = \mathbf{L}_\mathcal{C}^{-1}\Delta\mathbf{w}_\mathcal{C}^\top$ by solving the triangular system:

$$
\mathbf{L}_\mathcal{C}\mathbf{y} = \Delta\mathbf{w}_\mathcal{C}^\top. \tag{8}
$$

This approach reduces the operation to a strictly stable triangular solve, significantly improving numerical precision while maintaining the same computational complexity $\mathcal{O}(c^2)$ as matrix multiplication.

## 4.4. Spatial Alignment

Guided by the lookahead global objective derived in Eq. (7), we formulate the rounding problem as a subspace optimization task. To enable learnable rounding within the subspace, we incorporate the differentiable relaxation strategy from Adaround (Nagel et al., 2020). STLA aligns the optimization with the global error landscape by minimizing the Cholesky-transformed objective:

$$
\arg\min_{\mathbf{v}} \left\|\mathbf{L}_\mathcal{C}^{-1}\left(\mathbf{w}_\mathcal{C} - \widehat{\mathbf{w}}_\mathcal{C}(\mathbf{v})\right)^\top\right\|_F^2 + \lambda f_{reg}(\mathbf{v}). \tag{9}
$$

Where $\mathbf{w}_\mathcal{C}$ is the full-precision vector, $\widehat{\mathbf{w}}_\mathcal{C}$ is the quantized vector. By minimizing the $L_2$ norm of this projected error, we effectively solve the problem defined by the inverse of the inverse-Hessian submatrix without explicit inversion.

This lookahead global objective serves as a unified loss function across quantization parameter search, learning-based rounding, and compensation-based rounding components. It ensures mathematical consistency throughout the entire STLA framework.

## 4.5. Temporal Alignment

Once the spatial alignment $\widehat{\mathbf{w}}_\mathcal{C}(\mathbf{v})$ is established, we perform further cluster-wise optimization via temporal compensation. Traditional solvers like GPTQ perform weight compensation vector-by-vector. This sequential process introduces significant temporal misalignment, as quantization parameters (e.g., scale and zero-point) are typically derived from the original weight. As illustrated in Figure 1(a), the

weight distribution shifts continuously as preceding elements are compensated. The pre-determined quantization parameters become suboptimal, as the search space is no longer aligned with the updated weights. STLA resolves this by unifying the compensation granularity and quantization group, setting the cluster size $c$ equal to the group size. This enables the propagation of the aggregate error of the entire cluster to the remaining set $\mathcal{R}$ in a single update.

The optimal compensation update for the remaining weights $\mathbf{w}_\mathcal{R}$ is derived as:

$$
\mathbf{w}_\mathcal{R} \leftarrow \mathbf{w}_\mathcal{R} - (\mathbf{w}_\mathcal{C} - \widehat{\mathbf{w}}_\mathcal{C}(\mathbf{v})) \cdot \left((\mathbf{H}^{-1})_{\mathcal{C},\mathcal{C}}\right)^{-1} \cdot (\mathbf{H}^{-1})_{\mathcal{C},\mathcal{R}}. \tag{10}
$$

Here, $(\mathbf{H}^{-1})_{\mathcal{C},\mathcal{R}}$ denotes the rectangular slice capturing the correlations between the cluster and the remaining weights.

Crucially, to ensure computational efficiency, we adopt the numerical strategy from GPTQ. Rather than computing the explicit inverse $\left((\mathbf{H}^{-1})_{\mathcal{C},\mathcal{C}}\right)^{-1}$, we utilize the Cholesky decomposition of $\mathbf{H}^{-1}$. The compensation update can be expressed as:

$$
\mathbf{w}_\mathcal{R} \leftarrow \mathbf{w}_\mathcal{R} - (\mathbf{w}_\mathcal{C} - \widehat{\mathbf{w}}_\mathcal{C}(\mathbf{v})))(\mathbf{L}_\mathcal{C}^\top)^{-1}\mathbf{L}_{\mathcal{R},\mathcal{C}}^\top, \tag{11}
$$

where $\mathbf{L}_{\mathcal{R},\mathcal{C}}$ captures the cross-correlations in the Cholesky space. This formulation avoids double inversion and ensures robust numerical performance.

Directly computing the double inversion in Eq. (11) is numerically risky. To avoid this, we define the cluster error vector $\mathbf{e}_\mathcal{C} \in \mathbb{R}^c$ as:

$$
\mathbf{e}_\mathcal{C} = (\mathbf{w}_\mathcal{C} - \widehat{\mathbf{w}}_\mathcal{C}(\mathbf{v}))(\mathbf{L}_\mathcal{C}^\top)^{-1}. \tag{12}
$$

In practice, we compute $\mathbf{e}_\mathcal{C}$ by solving the following triangular linear system via back-substitution, which avoids explicit matrix inversion:

$$
\mathbf{L}_\mathcal{C}\mathbf{e}_\mathcal{C}^\top = (\mathbf{w}_\mathcal{C} - \widehat{\mathbf{w}}_\mathcal{C}(\mathbf{v}))^\top. \tag{13}
$$

This formulation ensures robust numerical performance by mapping the residual error directly into the Cholesky subspace. Assuming equal block and cluster sizes, the final inter-block compensation is then reduced to a simple matrix multiplication:

$$
\mathbf{w}_\mathcal{R} \leftarrow \mathbf{w}_\mathcal{R} - \mathbf{e}_\mathcal{C}\mathbf{L}_{\mathcal{R},\mathcal{C}}^\top. \tag{14}
$$

The STLA framework also supports cases where the block size exceeds the cluster size, as detailed in Appendix B.

## 4.6. Hessian-Guided Spin Clustering

The proposed coupled paradigm collocates learning and compensation, thereby reformulating the full-space learnable rounding objective into a subspace optimization task.

**Algorithm 1** SpinCluster: Sensitivity-Weighted Clustering

> **Input:** $\mathbf{H}$, cluster size $c$
> **Output:** $\mathbf{P}$ (ordered column index)
> $N \leftarrow$ number of columns in $\mathbf{W}$
> $\mathbf{D} \leftarrow \sqrt{\mathrm{diag}(\mathbf{H}_{\mathrm{head}})}$
> $\mathbf{D}^{-1} \leftarrow 1/\mathbf{D}$
> $\mathbf{S} \leftarrow \mathbf{D}^{-1}\mathbf{H}\mathbf{D}^{-1}$
> $\mathrm{diag}(\mathbf{S}) \leftarrow 0$
> $\mathcal{U} \leftarrow \{1, 2, \ldots, N\}$
> **while** $\mathcal{U} \neq \emptyset$ **do**
>    $k_0 \leftarrow \arg\max_{j \in \mathcal{U}} \mathbf{D}_j$
>    $C \leftarrow \{k_0\}$
>    $\mathbf{P} \leftarrow \{k_0\}$
>    $\mathcal{U} \leftarrow \mathcal{U} \setminus \{k_0\}$
>    $\phi \leftarrow \mathbf{S}_{k,:}$
>    **while** $|C| < c$ **and** $\mathcal{U} \neq \emptyset$ **do**
>       $k \leftarrow \arg\max_{j \in \mathcal{U}} (|\phi_j| \cdot \mathbf{D}_j)$
>       $q \leftarrow -\mathrm{sgn}(\phi_k)$
>       $C \leftarrow C \cup \{k\}$
>       $\mathbf{P} \leftarrow \mathbf{P} \cup \{k\}$
>       $\mathcal{U} \leftarrow \mathcal{U} \setminus \{k\}$
>       $\phi \leftarrow \phi + q \cdot \mathbf{S}_{k,:}$
>    **end while**
>    $\mathbf{P} \leftarrow \mathbf{P} \cup \{C\}$
> **end while**
> $\mathbf{P}$

The efficacy of this learning-based rounding relies on leveraging intra-cluster weight correlations to minimize the reconstruction loss $\mathcal{L}$, decomposed as follows:

$$\mathcal{L} \approx \sum_i \mathbf{H}_{ii} \Delta w_i^2 + \sum_{i \neq j} \mathbf{H}_{ij} \Delta w_i \Delta w_j. \quad (15)$$

The cross-term $\mathbf{H}_{ij}\Delta w_i \Delta w_j$ enables error cancellation. For strongly coupled weights ($|\mathbf{H}_{ij}| \gg 0$), the total loss is minimized by aligning the signs of $\Delta w_i$ and $\Delta w_j$ to neutralize their joint contribution. Learning-based rounding operates by identifying these optimal rounding directions, frequently referred to as spins. However, standard methods process weights in their natural order, often separating highly correlated pairs across different clusters and thereby sacrificing potential error cancellation. To address this, we propose spin clustering (SpinCluster), which reorders weights into clusters that encapsulate dense correlation subgraphs, maximizing the optimization efficacy of the learning-based adaptive rounding solver.

For each cluster, SpinCluster utilizes diagonal Hessian information to quantify weight sensitivity and off-diagonal information to capture inter-weight correlations, as illustrated in Figure 2. The SpinCluster algorithm is summarized in Algorithm 1. We first isolate the interaction topology by

defining a sensitivity matrix $\mathbf{D}$ and a normalized correlation matrix $\mathbf{S}$:

$$\mathbf{D} = \mathrm{diag}\left(\sqrt{\mathbf{H}_{11}}, \ldots, \sqrt{\mathbf{H}_{NN}}\right), \quad \mathbf{S} = \mathbf{D}^{-1}\mathbf{H}\mathbf{D}^{-1}, \quad (16)$$

where $\mathbf{D}$ represents individual sensitivity and $\mathbf{S}$ captures pairwise connectivity, with $\mathrm{diag}(\mathbf{S}) = 0$. Each cluster is initialized with a seed node $k_0$:

$$k_0 = \arg\max_{j \in \mathcal{U}} \mathbf{D}_{jj}, \quad (17)$$

where $\mathcal{U}$ denotes the unvisited indices. We establish an initial interaction vector $\phi^{(0)} = \mathbf{S}_{k_0,:}$. Here, $\mathbf{S}_{k_0,:}$ denotes the $k_0$-th row of $\mathbf{S}$. The cluster is then iteratively expanded by selecting $k_{t+1}$ and updating the field:

$$k_{t+1} = \arg\max_{j \in \mathcal{U}} \left(|\phi_j^{(t)}| \cdot \mathbf{D}_{jj}\right),$$
$$\phi^{(t+1)} = \phi^{(t)} - \mathrm{sgn}(\phi_{k_{t+1}}^{(t)}) \cdot \mathbf{S}_{k_{t+1},:}. \quad (18)$$

This greedy update assigns opposing spins to weights with strong mutual interactions, maximizing intra-cluster destructive interference and providing a near-optimal initial state for the subsequent solver.

### 4.7. Complexity Analysis

We provide a rigorous analysis of the layer-wise computational complexity of STLA, demonstrating that it achieves superior quantization accuracy while reducing computational cost compared to the standard decoupled paradigm. Let $d_{in}$ and $d_{out}$ denote the input and output dimensions.

**SpinCluster Overhead.** The reordering process in Algorithm 1 operates on the normalized Hessian $\mathbf{S} \in \mathbb{R}^{d_{in} \times d_{in}}$. Within each cluster, the first weight selection requires $d_{in}$ comparisons, and subsequent search steps involve updating the potential vector $\phi \in \mathbb{R}^{d_{in}}$ and searching for the maximum score. This results in $\mathcal{O}(c \cdot d_{in})$ operations per cluster. Aggregating over all $d_{in}/c$ clusters, the total complexity is:

$$\mathcal{K}_{cluster} = \frac{d_{in}}{c} \cdot \mathcal{O}(c \cdot d_{in}) = \mathcal{O}(d_{in}^2). \quad (19)$$

**Learning-based Rounding Efficiency.** While collocating learning and compensation might intuitively seem to increase computational overhead, our analysis reveals this is not true. Learnable rounding within each cluster involves $m$ iterations of gradient descent, where each iteration comprises: (1) a forward pass requiring a triangular solve $\mathbf{L}_{\mathcal{C}}\mathbf{x} = \Delta\mathbf{w}^\top$ with cost $\mathcal{O}(c^2)$; and (2) a backward pass to compute gradients with respect to the rounding parameters, which similarly involves a transposed triangular solve and vector operations scaling as $\mathcal{O}(c^2)$. The total computational cost for a layer is:

$$\mathcal{K}_{learn} = \frac{d_{in}}{c} \cdot d_{out} \cdot \mathcal{O}(m \cdot (c_{fwd}^2 + c_{bwd}^2)),$$
$$= \mathcal{O}(m \cdot d_{in} \cdot d_{out} \cdot c). \quad (20)$$

*Table 1.* Ablation results evaluating the contributions of the lookahead global objective, the coupled spatiotemporal alignment optimization, and the SpinCluster strategy, evaluated in terms of PPL on C4 and WikiText-2 (Wkt-2) dataset, and GPU runtime.

| GRAN. | GROUP SIZE | COUPLED | CLUSTER ORDER | LOSS | OPT-125M | | | LLAMA2-7B | | |
|---|---|---|---|---|---|---|---|---|---|---|
| | | | | | C4 | WKT-2 | TIME | C4 | WKT-2 | TIME |
| CHANNEL | - | × | × | GLOBAL | 31.21 | 34.52 | 109.3 S | 8.42 | 6.51 | 2.3 HR |
| CHANNEL | - | ✓ | × | LOCAL | 32.63 | 35.82 | 74.7 S | 8.47 | 6.49 | 36.4 MIN |
| CHANNEL | - | ✓ | × | GLOBAL | 30.69 | 33.45 | 76.5 S | 8.30 | 6.36 | 38.6 MIN |
| GROUP | 256 | × | × | GLOBAL | 30.33 | 33.59 | 116.4 S | 8.09 | 6.19 | 2.3 HR |
| GROUP | 256 | ✓ | × | LOCAL | 31.30 | 35.86 | 75.2 S | 8.14 | 6.11 | 32.8 MIN |
| GROUP | 256 | ✓ | × | GLOBAL | 29.86 | 32.00 | 76.8 S | 8.02 | 6.44 | 32.8 MIN |
| GROUP | 256 | ✓ | SPIN | LOCAL | 30.51 | 33.39 | 89.4 S | 8.10 | 6.17 | 39.7 MIN |
| GROUP | 256 | ✓ | SPIN | GLOBAL | 29.26 | 31.01 | 89.8 S | 7.99 | 6.09 | 40.6 MIN |

Notably, reformulating full-space learnable rounding as a subspace optimization task not only reduces the per-iteration computational cost from $\mathcal{O}(d_{\text{in}}^2 \cdot d_{out})$ to $\mathcal{O}(d_{\text{in}} \cdot d_{out} \cdot c)$ but also significantly lowers the number of iterations $m$ required for convergence. This is attributed to the simplified optimization landscape within the subspace, which mitigates the complexity of finding global optima in high-dimensional settings. As exemplified in Figure 1, subspace learning converges in just 200 iterations, compared to 2000 for the full-space approach.

**Compensation Efficiency.** The compensation update propagates joint errors via the cross-correlation slice $(\mathbf{H}^{-1})_{\mathcal{C},\mathcal{R}}$. The total computational cost is:

$$\mathcal{K}_{comp} = \sum_{i=1}^{d_{in}/c} \mathcal{O}(d_{out} \cdot c \cdot (d_{in} - i \cdot c)) = \mathcal{O}(d_{out} \cdot d_{in}^2). \tag{21}$$

**Comparison with Decoupled Paradigm.** For each layer, the total computational complexity of STLA is:

$$\mathcal{K}_{STLA} = \mathcal{O}(d_{in}^2) + \mathcal{O}(m \cdot d_{in} \cdot d_{out} \cdot c) + \mathcal{O}(d_{out} \cdot d_{in}^2),$$
$$\approx \mathcal{O}(m \cdot d_{in} \cdot d_{out} \cdot c) + \mathcal{O}(d_{out} \cdot d_{in}^2). \tag{22}$$

In contrast, the decoupled paradigm necessitates a full layer-wise learnable rounding phase of $t$ iterations following compensation, yielding a complexity of:

$$\mathcal{K}_{decoupled} = \mathcal{O}(t \cdot d_{in}^2 \cdot d_{out}) + \mathcal{O}(d_{out} \cdot d_{in}^2). \tag{23}$$

While maintaining equivalent compensation complexity, STLA significantly reduces the total cost compared to the decoupled paradigm, given that $m \cdot c \ll t \cdot d_{in}$.

## 5. Experiments

We assess the performance of STLA across various models, including a rigorous ablation study to validate the proposed spatial and temporal alignment mechanisms. STLA is also benchmarked against state-of-the-art (SOTA) PTQ baselines, reporting results on both PPL and zero-shot accuracy. Additional experiments and results are detailed in the Appendix.

### 5.1. Experimental Setup

We apply our method to the pre-trained FP16 weights of two prominent model families: OPT (Zhang et al., 2022) and LLaMA (Touvron et al., 2023). Our experiments focus on weight-only quantization, targeting 3-bit and 2-bit group-wise PTQ configurations. Unless explicitly stated otherwise, the group, block, and cluster sizes are set to 256. To verify the robustness of STLA, we also evaluate it across diverse granularities. This includes channel-wise configurations and varied group/block/cluster sizes (e.g., $c = 128$ vs. $c = 256$), which offer a finer trade-off between accuracy and hardware efficiency. Details are provided in Appendix C.

For the calibration process, we randomly sample 128 segments from the C4 (Raffel et al., 2020) training set, each with a sequence length of 2,048 tokens, consistent with the protocols established in prior second-order PTQ literature. To assess the linguistic capabilities and reasoning performance of the quantized models, we report the PPL on the WikiText-2 (Merity et al., 2017) and C4 test set. Furthermore, we conduct a comprehensive evaluation on zero-shot benchmarks, including PIQA (Bisk et al., 2020), ARC (Easy and Challenge) (Clark et al., 2018), HellaSwag (Zellers et al., 2019), WinoGrande (Sakaguchi et al., 2021) and MMLU (Hendrycks et al., 2021). All quantization and optimization procedures are executed on a single NVIDIA RTX 5090 GPU (32GB).

### 5.2. Ablation Study

Our ablation study, detailed in Table 1, validates the spatial and temporal alignment components within the STLA framework. For learning-based rounding, we utilize the Adam optimizer with a CosineAnnealingLR scheduler. In STLA, the learning rate is initialized at 1.1 for OPT-125M and 0.5 for LLaMA2-7B, subsequently decaying to 30% of its initial value over 200 iterations per cluster. In contrast, the decoupled paradigm necessitates a significantly longer optimization phase, requiring 2000 iterations with a fixed learning rate of 0.015 to achieve comparable convergence. This disparity highlights the superior efficiency of our sub-

*Table 2.* Performance (PPL) of 3-bit group-wise (group size is set to 256) weight quantization of STLA and existing PTQ methods on LLM models. (Calibration data from C4)

| Method | OPT-125M | | OPT-1.3B | | OPT-6.7B | | LLaMA2-7B | | LLaMA3-8B | |
|---|---|---|---|---|---|---|---|---|---|---|
| | WKT-2(↓) | C4(↓) | WKT-2(↓) | C4(↓) | WKT-2(↓) | C4(↓) | WKT-2(↓) | C4(↓) | WKT-2(↓) | C4(↓) |
| FP16 | 27.65 | 26.56 | 14.62 | 16.07 | 10.86 | 12.71 | 5.47 | 7.26 | 6.14 | 9.45 |
| GPTQ | 43.89 | 37.61 | 16.72 | 17.87 | 12.68 | 14.78 | 6.54 | 8.42 | 8.23 | 12.80 |
| AWQ | 40.62 | 35.63 | 16.17 | 18.04 | 11.31 | 13.29 | 6.44 | 8.49 | 8.59 | 13.42 |
| GPTAQ | 42.52 | 36.17 | 16.57 | 17.88 | 12.90 | 14.94 | 6.24 | 8.22 | 8.00 | 12.58 |
| STLA | **31.01** | **29.26** | **15.24** | **17.12** | **11.06** | **13.17** | **6.09** | **7.99** | **7.82** | **11.88** |

*Table 3.* Zero-shot accuracy of 3-bit group-wise (group size = 256) weight quantization of STLA and existing PTQ methods on LLaMA2-7B. (Calibration data from C4)

| Method | PIQA | ARC-E | ARC-C | HellaSwag | WinoGrande | MMLU | Avg(↑) |
|---|---|---|---|---|---|---|---|
| FP16 | 78.78 | 73.82 | 45.05 | 76.15 | 69.38 | 41.76 | 64.16 |
| GPTQ (Frantar et al., 2023) | 76.50 | 67.80 | 41.55 | 70.24 | 68.43 | 34.08 | 59.77 |
| AWQ (Lin et al., 2024) | 76.22 | 68.39 | 40.78 | 70.60 | 65.04 | 34.16 | 59.20 |
| GPTAQ (Li et al., 2025) | 77.48 | 69.23 | 41.81 | 71.76 | 68.67 | 34.66 | 60.60 |
| STLA | **77.64** | **70.96** | **42.32** | **72.61** | **69.22** | **38.52** | **61.88** |

space learning approach. Throughout all experiments, the rounding loss weight is maintained at 1.

We evaluate the impact of quantization granularity, the coupled paradigm, clustering strategy, and the optimization objective in Table 1. For per-channel PTQ, although decoupled methods inherently utilize a global objective, our results demonstrate that coupling the learning-based rounding process with compensation significantly reduces perplexity. This improvement is largely attributed to temporal alignment, which resolves the conflict associated with the RTN assumption in the decoupled paradigm. Within the coupled framework, the global objective generally outperforms the local loss, improving performance across most evaluated settings. Although the individual benefits are not strictly uniform across all configurations, the lookahead global objective achieves peak performance when combined synergistically with the other proposed components in STLA. Group-wise quantization achieves higher accuracy than channel-wise PTQ, particularly when aligned with the STLA cluster size. The integration of SpinCluster further minimizes reconstruction error by maximizing intra cluster error cancellation. Collectively, these results demonstrate that STLA achieves the best accuracy while maintaining high efficiency.

### 5.3. PPL Evaluation

We benchmark our framework against GPTQ, AWQ, and GPTAQ as these methods provide native support for group-wise PTQ. The PPL results for WikiText 2 and C4 datasets are summarized in Table 2 and Table 4.

In 3 bit configuration, STLA consistently delivers the lowest PPL across all models. For the OPT-125M model, our approach reduces the WikiText-2 PPL to 31.01 compared to 43.89 in GPTQ, 42.52 in GPTAQ and 40.62 in AWQ, demonstrating a significant improvement in reconstruction fidelity. This trend remains consistent for larger architectures, such as LLaMA2-7B and LLaMA3-8B, where STLA achieves PPLs of 6.09 and 7.82, respectively.

These advantages become even more pronounced in the extremely low-bit regime. As shown in Table 4, while standard methods suffer from severe performance degradation at 2-bit PTQ, STLA maintains competitive accuracy and exhibits superior robustness. Notably, for LLaMA2-7B on WikiText-2, STLA achieves a PPL of 10.02, which is significantly lower than the 30.10 recorded for GPTQ. These results indicate that through spatial and temporal alignment optimization, STLA effectively prevents the accumulation of misaligned rounding errors that typically lead to model failure at low bit-widths.

### 5.4. Zero-Shot Accuracy Evaluation

We also evaluated the performance of STLA through zero shot accuracy on LLaMA2-7B as shown in Table 3 and Table 5. At 3 bit PTQ, STLA achieves an average accuracy of 61.88%, significantly outperforming other methods and representing the minimal gap to the FP16 baseline. The robustness of STLA is even more evident in the 2 bit regime where standard methods exhibit catastrophic failure. While AWQ and GPTQ drop below 40%, STLA maintains a competitive 51.55%.

*Table 4.* Performance (PPL) of 2-bit group-wise (group size is set to 256) weight quantization of STLA and existing PTQ methods on LLM models. (Calibration data from C4)

| METHOD | OPT-125M | | OPT-1.3B | | OPT-6.7B | | LLaMA2-7B | |
|---|---|---|---|---|---|---|---|---|
| | WIKITEXT-2($\downarrow$) | C4($\downarrow$) | WIKITEXT-2($\downarrow$) | C4($\downarrow$) | WIKITEXT-2($\downarrow$) | C4($\downarrow$) | WIKITEXT-2($\downarrow$) | C4($\downarrow$) |
| FP16 | 27.65 | 26.56 | 14.62 | 16.07 | 10.86 | 12.71 | 5.47 | 7.26 |
| GPTQ | 282.64 | 161.18 | 57.17 | 59.00 | 36.55 | 48.81 | 30.10 | 41.03 |
| AWQ | 819.27 | 502.98 | 139.34 | 117.01 | 21.73 | 24.45 | 2.29E5 | 1.73E5 |
| GPTAQ | 189.80 | 108.10 | 43.76 | 41.74 | 28.44 | 29.49 | 18.86 | 16.27 |
| STLA | **62.33** | **50.69** | **25.31** | **26.89** | **15.41** | **17.75** | **10.02** | **12.31** |

*Table 5.* Zero-shot accuracy of 2-bit group-wise (group size = 256) weight quantization of STLA and existing PTQ methods on LLaMA2-7B. (Calibration data from C4)

| METHOD | PIQA | ARC-E | ARC-C | HELLASWAG | WINOGRANDE | MMLU | AVG($\uparrow$) |
|---|---|---|---|---|---|---|---|
| FP16 | 78.78 | 73.82 | 45.05 | 76.15 | 69.38 | 41.76 | 64.16 |
| GPTQ (FRANTAR ET AL., 2023) | 60.77 | 39.27 | 24.06 | 39.31 | 52.64 | 22.94 | 39.83 |
| AWQ (LIN ET AL., 2024) | 50.11 | 26.47 | 27.30 | 26.11 | 49.09 | 25.51 | 34.10 |
| GPTAQ (LI ET AL., 2025) | 65.13 | 44.70 | 26.71 | 46.65 | 57.06 | 22.39 | 43.77 |
| STLA | **74.27** | **55.89** | **31.66** | **57.64** | **62.19** | **27.66** | **51.55** |

*Table 6.* Perplexity comparison of STLA against baseline quantization methods integrated with rotation techniques, including QuaRot and SpinQuant, on LLaMA3-8B and LLaMA2-7B under W3A4 and W2A4 settings.

| METHOD | W3A4 | | | | W2A4 | | | |
|---|---|---|---|---|---|---|---|---|
| | LLaMA3-8B | | LLaMA2-7B | | LLaMA3-8B | | LLaMA2-7B | |
| | WIKITEXT-2 | C4 | WIKITEXT-2 | C4 | WIKITEXT-2 | C4 | WIKITEXT-2 | C4 |
| QUAROT+GPTQ | 10.82 | 16.62 | 7.10 | 9.58 | 49.44 | 60.86 | 42.21 | 46.86 |
| QUAROT+GPTAQ | 9.72 | 15.00 | 6.55 | 8.77 | 33.14 | 37.34 | 16.90 | 18.71 |
| QUAROT+STLA | **9.27** | **14.35** | **6.38** | **8.51** | **25.72** | **29.91** | **11.42** | **13.62** |
| SPINQUANT+GPTQ | 9.61 | 13.38 | 6.91 | 8.62 | 40.71 | 46.89 | 45.60 | 41.64 |
| SPINQUANT+GPTAQ | 9.30 | 12.87 | 6.51 | 8.13 | 27.18 | 29.19 | 19.43 | 19.15 |
| SPINQUANT+STLA | **8.91** | **12.44** | **6.33** | **7.94** | **21.20** | **22.79** | **11.69** | **12.76** |

## 5.5. Compatibility with Rotation-Based Quantization

STLA is fully compatible with rotation-based methods. We evaluate STLA in combination with rotation-based quantization techniques, including QuaRot and SpinQuant. As shown in Table 6, STLA consistently improves upon GPTQ and GPTAQ within the same rotation pipeline, demonstrating its compatibility with rotation-based preprocessing and confirming its efficacy in realistic deployment scenarios.

## 6. Conclusion

This paper introduces STLA, a unified framework designed to address spatial and temporal misalignments in existing PTQ methods. STLA resolves temporal conflicts by collocating the learning and compensation phases, while achieving spatial alignment through a unified global objective. Furthermore, STLA introduces a Hessian-guided spin clus-

tering strategy to enhance cluster-wise error cancellation. This framework establishes a rigorous mathematical foundation to capture and neutralize second-order reconstruction errors with minimal computational overhead. Experiments demonstrate that STLA consistently achieves superior performance in the 3-bit and 2-bit regimes. These findings validate STLA as an efficient solution for the deployment of LLMs in edge scenarios.

## Acknowledgements

This work was supported in part by the Science and Technology Development Fund, Macau SAR (FDCT) under Grant 0001/2025/NRP and Grant 0020/2025/ITP1; in part by the University of Macau under Grant MYRG-GRG2025-00283-IME, Grant SRG2024-00055-IME, and Grant UMDF-TISF/2025/012/IME.

## Impact Statement

This paper presents work whose goal is to advance the field of Model Compression. There are many potential societal consequences of our work, none which we feel must be specifically highlighted here.

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

# A. Theoretical Derivation

## A.1. Loss Funtion

We derive the global alignment objective and demonstrate its relationship to the standard local metric. We show that the local Hessian metric is a specific case of the global objective constrained by static remaining weights, whereas our formulation generalizes OBQ to high-dimensional subspaces.

### A.1.1. PROBLEM SETUP

Consider the quantization of a layer with weight matrix $\mathbf{W}$. The Taylor expansion of the loss function with respect to the perturbation $\Delta \mathbf{w}$ is approximated by:

$$\mathcal{L}(\Delta \mathbf{w}) = \Delta \mathbf{w} \mathbf{H} \Delta \mathbf{w}^\top, \tag{24}$$

where $\mathbf{H}$ is the Hessian matrix. We partition the weights into the current cluster $\mathcal{C}$ and the remaining weights $\mathcal{R}$. Accordingly, the perturbation vector and Hessian are partitioned as:

$$\Delta \mathbf{w} = \begin{bmatrix} \Delta \mathbf{w}_\mathcal{C} & \Delta \mathbf{w}_\mathcal{R} \end{bmatrix}, \quad \mathbf{H} = \begin{bmatrix} \mathbf{H}_{\mathcal{C},\mathcal{C}} & \mathbf{H}_{\mathcal{C},\mathcal{R}} \\ \mathbf{H}_{\mathcal{R},\mathcal{C}} & \mathbf{H}_{\mathcal{R},\mathcal{R}} \end{bmatrix}. \tag{25}$$

The expanded error term is:

$$\mathcal{L} = \Delta \mathbf{w}_\mathcal{C} \mathbf{H}_{\mathcal{C},\mathcal{C}} \Delta \mathbf{w}_\mathcal{C}^\top + 2 \Delta \mathbf{w}_\mathcal{C} \mathbf{H}_{\mathcal{C},\mathcal{R}} \Delta \mathbf{w}_\mathcal{R}^\top + \Delta \mathbf{w}_\mathcal{R} \mathbf{H}_{\mathcal{R},\mathcal{R}} \Delta \mathbf{w}_\mathcal{R}^\top. \tag{26}$$

### A.1.2. DERIVATION OF THE LOCAL OBJECTIVE

Standard solvers assume the optimization of the current cluster is independent of the remaining weights, implying $\Delta \mathbf{w}_\mathcal{R} = \mathbf{0}$. Under this constraint, Eq. (26) reduces to:

$$\mathcal{L}_{local}(\Delta \mathbf{w}_\mathcal{C}) = \Delta \mathbf{w}_\mathcal{C} \mathbf{H}_{\mathcal{C},\mathcal{C}} \Delta \mathbf{w}_\mathcal{C}^\top. \tag{27}$$

This formulation neglects the cross-term $\mathbf{H}_{\mathcal{C},\mathcal{R}}$, isolating the current subspace from global optimization.

### A.1.3. DERIVATION OF THE GLOBAL OBJECTIVE

In sequential frameworks, $\mathbf{w}_\mathcal{R}$ is updated in subsequent steps to compensate for the error introduced by $\Delta \mathbf{w}_\mathcal{C}$. The objective is to find the optimal $\Delta \mathbf{w}_\mathcal{C}$ conditioned on the optimal adjustment of $\Delta \mathbf{w}_\mathcal{R}$.

Minimizing Eq. (26) with respect to $\Delta \mathbf{w}_\mathcal{R}$ yields the optimal compensation update $\Delta \mathbf{w}_\mathcal{R}^*$:

$$\frac{\partial \mathcal{L}}{\partial \Delta \mathbf{w}_\mathcal{R}} = 0 \implies \Delta \mathbf{w}_\mathcal{R}^* = -\Delta \mathbf{w}_\mathcal{C} \mathbf{H}_{\mathcal{C},\mathcal{R}} (\mathbf{H}_{\mathcal{R},\mathcal{R}})^{-1}. \tag{28}$$

Substituting $\Delta \mathbf{w}_\mathcal{R}^*$ back into Eq. (26):

$$\mathcal{L}_{global}(\Delta \mathbf{w}_\mathcal{C}) = \Delta \mathbf{w}_\mathcal{C} \mathbf{H}_{\mathcal{C},\mathcal{C}} \Delta \mathbf{w}_\mathcal{C}^\top + \Delta \mathbf{w}_\mathcal{C} \mathbf{H}_{\mathcal{C},\mathcal{R}} (\Delta \mathbf{w}_\mathcal{R}^*)^\top, \tag{29}$$

$$= \Delta \mathbf{w}_\mathcal{C} \left( \mathbf{H}_{\mathcal{C},\mathcal{C}} - \mathbf{H}_{\mathcal{C},\mathcal{R}} (\mathbf{H}_{\mathcal{R},\mathcal{R}})^{-1} \mathbf{H}_{\mathcal{R},\mathcal{C}} \right) \Delta \mathbf{w}_\mathcal{C}^\top. \tag{30}$$

The term in parentheses is the Schur Complement of $\mathbf{H}_{\mathcal{R},\mathcal{R}}$ in $\mathbf{H}$, representing the effective curvature of subspace $\mathcal{C}$ after marginalizing $\mathcal{R}$.

Computing the Schur Complement directly involves computationally expensive inversions. We utilize the block matrix inversion lemma. For the inverse Hessian $\mathbf{H}^{-1}$, the top-left block corresponding to $\mathcal{C}$ is:

$$(\mathbf{H}^{-1})_{\mathcal{C},\mathcal{C}} = \left( \mathbf{H}_{\mathcal{C},\mathcal{C}} - \mathbf{H}_{\mathcal{C},\mathcal{R}} (\mathbf{H}_{\mathcal{R},\mathcal{R}})^{-1} \mathbf{H}_{\mathcal{R},\mathcal{C}} \right)^{-1}. \tag{31}$$

Therefore, the global objective simplifies to:

$$\mathcal{L}_{global}(\Delta \mathbf{w}_\mathcal{C}) = \Delta \mathbf{w}_\mathcal{C} \left( (\mathbf{H}^{-1})_{\mathcal{C},\mathcal{C}} \right)^{-1} \Delta \mathbf{w}_\mathcal{C}^\top. \tag{32}$$

## A.2. Temporal Compensation

This section provides a rigorous derivation of the weight update formula from the first principles of constrained optimization, independent of the previous sections.

### A.2.1. PROBLEM SETUP

Consider the quantization of a weight cluster $\mathcal{C}$ with size $c$. We seek a full perturbation row-vector $\Delta\mathbf{w} \in \mathbb{R}^{1 \times N}$ that minimizes the total error $\mathcal{L} = \frac{1}{2}\Delta\mathbf{w}\mathbf{H}\Delta\mathbf{w}^{\top}$, subject to the constraint that the weights in $\mathcal{C}$ must match their quantized values $\text{quant}(\mathbf{w}_{\mathcal{C}})$.

We define the selection matrix $\mathbf{E}_{\mathcal{C}} \in \{0,1\}^{c \times N}$, where each row $k$ is a standard basis vector $\mathbf{e}_{i_k}^{\top}$ for $i_k \in \mathcal{C}$. The constraint is written as:

$$\mathbf{E}_{\mathcal{C}}\Delta\mathbf{w}^{\top} = (\widehat{\mathbf{w}}_{\mathcal{C}} - \mathbf{w}_{\mathcal{C}})^{\top}. \tag{33}$$

We define the Lagrangian $J$ with a multiplier vector $\boldsymbol{\lambda} \in \mathbb{R}^{c \times 1}$ as:

$$J(\Delta\mathbf{w}, \boldsymbol{\lambda}) = \frac{1}{2}\Delta\mathbf{w}\mathbf{H}\Delta\mathbf{w}^{\top} + \boldsymbol{\lambda}^{\top}\left(\mathbf{E}_{\mathcal{C}}\Delta\mathbf{w}^{\top} - (\widehat{\mathbf{w}}_{\mathcal{C}} - \mathbf{w}_{\mathcal{C}})^{\top}\right). \tag{34}$$

### A.2.2. SOLVING FOR THE GLOBAL PERTURBATION

Taking the gradient with respect to $\Delta\mathbf{w}$ and setting it to zero:

$$\nabla_{\Delta\mathbf{w}}J = \Delta\mathbf{w}\mathbf{H} + \boldsymbol{\lambda}^{\top}\mathbf{E}_{\mathcal{C}} = \mathbf{0} \implies \Delta\mathbf{w}^{\top} = -\mathbf{H}^{-1}\mathbf{E}_{\mathcal{C}}^{\top}\boldsymbol{\lambda}. \tag{35}$$

Substituting this stationary point into the constraint equation:

$$\mathbf{E}_{\mathcal{C}}(-\mathbf{H}^{-1}\mathbf{E}_{\mathcal{C}}^{\top}\boldsymbol{\lambda}) = (\widehat{\mathbf{w}}_{\mathcal{C}} - \mathbf{w}_{\mathcal{C}})^{\top}. \tag{36}$$

Recognizing that $\mathbf{E}_{\mathcal{C}}\mathbf{H}^{-1}\mathbf{E}_{\mathcal{C}}^{\top}$ is precisely the $c \times c$ sub-block of the inverse Hessian $(\mathbf{H}^{-1})_{\mathcal{C},\mathcal{C}}$, we solve for the multipliers:

$$\boldsymbol{\lambda} = -\left((\mathbf{H}^{-1})_{\mathcal{C},\mathcal{C}}\right)^{-1}(\widehat{\mathbf{w}}_{\mathcal{C}} - \mathbf{w}_{\mathcal{C}})^{\top}. \tag{37}$$

### A.2.3. EXTRACTION OF THE REMAINING WEIGHTS UPDATE

To find the specific update for the remaining weights $\mathbf{w}_{\mathcal{R}}$, we define the selection matrix $\mathbf{E}_{\mathcal{R}} \in \{0,1\}^{(N-c) \times N}$ for indices $j \in \mathcal{R}$. The perturbation for these weights is extracted as:

$$\Delta\mathbf{w}_{\mathcal{R}}^{\top} = \mathbf{E}_{\mathcal{R}}\Delta\mathbf{w}^{\top} = -\mathbf{E}_{\mathcal{R}}\mathbf{H}^{-1}\mathbf{E}_{\mathcal{C}}^{\top}\boldsymbol{\lambda}. \tag{38}$$

By substituting $\boldsymbol{\lambda}$, we have:

$$\Delta\mathbf{w}_{\mathcal{R}}^{\top} = (\mathbf{H}^{-1})_{\mathcal{R},\mathcal{C}}\left((\mathbf{H}^{-1})_{\mathcal{C},\mathcal{C}}\right)^{-1}(\widehat{\mathbf{w}}_{\mathcal{C}} - \mathbf{w}_{\mathcal{C}})^{\top}. \tag{39}$$

To ensure numerical stability and computational efficiency, we introduce the Cholesky decomposition of the inverse Hessian $\mathbf{H}^{-1} = \mathbf{L}\mathbf{L}^{\top}$, where $\mathbf{L}$ is lower triangular:

$$\mathbf{L} = \begin{bmatrix} \mathbf{L}_{\mathcal{C}} & \mathbf{0} \\ \mathbf{L}_{\mathcal{R},\mathcal{C}} & \mathbf{L}_{\mathcal{R}} \end{bmatrix}. \tag{40}$$

The sub-blocks of the inverse Hessian can be expressed as $(\mathbf{H}^{-1})_{\mathcal{C},\mathcal{C}} = \mathbf{L}_{\mathcal{C}}\mathbf{L}_{\mathcal{C}}^{\top}$ and $(\mathbf{H}^{-1})_{\mathcal{R},\mathcal{C}} = \mathbf{L}_{\mathcal{R},\mathcal{C}}\mathbf{L}_{\mathcal{C}}^{\top}$. Substituting these into the perturbation formula:

$$\Delta\mathbf{w}_{\mathcal{R}}^{\top} = (\mathbf{L}_{\mathcal{R},\mathcal{C}}\mathbf{L}_{\mathcal{C}}^{\top})(\mathbf{L}_{\mathcal{C}}\mathbf{L}_{\mathcal{C}}^{\top})^{-1}(\widehat{\mathbf{w}}_{\mathcal{C}} - \mathbf{w}_{\mathcal{C}})^{\top}, \tag{41}$$

$$= \mathbf{L}_{\mathcal{R},\mathcal{C}}\mathbf{L}_{\mathcal{C}}^{\top}(\mathbf{L}_{\mathcal{C}}^{\top})^{-1}\mathbf{L}_{\mathcal{C}}^{-1}(\widehat{\mathbf{w}}_{\mathcal{C}} - \mathbf{w}_{\mathcal{C}})^{\top}, \tag{42}$$

$$= \mathbf{L}_{\mathcal{R},\mathcal{C}}\mathbf{L}_{\mathcal{C}}^{-1}(\widehat{\mathbf{w}}_{\mathcal{C}} - \mathbf{w}_{\mathcal{C}})^{\top}. \tag{43}$$

Transposing the vector back to row-notation and substituting the residual $(\widehat{\mathbf{w}}_{\mathcal{C}} - \mathbf{w}_{\mathcal{C}}) = -(\mathbf{w}_{\mathcal{C}} - \widehat{\mathbf{w}}_{\mathcal{C}})$, we obtain the final update rule:

$$\mathbf{w}_{\mathcal{R}} \leftarrow \mathbf{w}_{\mathcal{R}} - (\mathbf{w}_{\mathcal{C}} - \widehat{\mathbf{w}}_{\mathcal{C}})(\mathbf{L}_{\mathcal{C}}^{\top})^{-1}\mathbf{L}_{\mathcal{R},\mathcal{C}}^{\top}. \tag{44}$$

This completes the derivation of the cluster-wise temporal compensation using Cholesky factors.

### A.3. Bound Analysis of SpinCluster

Although SpinCluster adopts a greedy clustering strategy, its design is directly motivated by the second-order quantization objective. Let

$$L_{\text{full}}(\Delta\mathbf{w}) = \sum_i H_{ii}\Delta w_i^2 + \sum_{i\neq j} H_{ij}\Delta w_i\Delta w_j \tag{45}$$

denote the full quadratic objective, where $H$ is the Hessian matrix and $\Delta\mathbf{w}$ is the quantization-induced weight perturbation. Let $L_{\text{cluster}}(\Delta\mathbf{w})$ denote the cluster-restricted objective that preserves only the within-cluster Hessian interactions. If $\mathcal{E}_{\text{cut}}$ denotes the set of cross-cluster index pairs, the difference between the full objective and the cluster-restricted objective is given by the omitted cross-cluster interactions:

$$L_{\text{full}}(\Delta\mathbf{w}) - L_{\text{cluster}}(\Delta\mathbf{w}) = \sum_{(i,j)\in\mathcal{E}_{\text{cut}}} H_{ij}\Delta w_i\Delta w_j. \tag{46}$$

Taking the absolute value gives

$$|L_{\text{full}}(\Delta\mathbf{w}) - L_{\text{cluster}}(\Delta\mathbf{w})| \le \sum_{(i,j)\in\mathcal{E}_{\text{cut}}} |H_{ij}|\,|\Delta w_i|\,|\Delta w_j|. \tag{47}$$

Assuming the perturbation of each weight is bounded by $|\Delta w_i| \le \epsilon$, we further obtain

$$|L_{\text{full}}(\Delta\mathbf{w}) - L_{\text{cluster}}(\Delta\mathbf{w})| \le \epsilon^2 \sum_{(i,j)\in\mathcal{E}_{\text{cut}}} |H_{ij}|. \tag{48}$$

This bound shows that the approximation error introduced by cluster-wise optimization is controlled by the magnitude of the omitted cross-cluster Hessian interactions. Therefore, grouping weights with strong Hessian correlations into the same cluster reduces the cut interaction term $\sum_{(i,j)\in\mathcal{E}_{\text{cut}}} |H_{ij}|$, allowing the cluster-restricted objective to better preserve the structure of the full second-order objective. This provides a theoretical justification for the Hessian-guided design of SpinCluster.

## B. Definition of Layer, Channel, Group, Block and Cluster

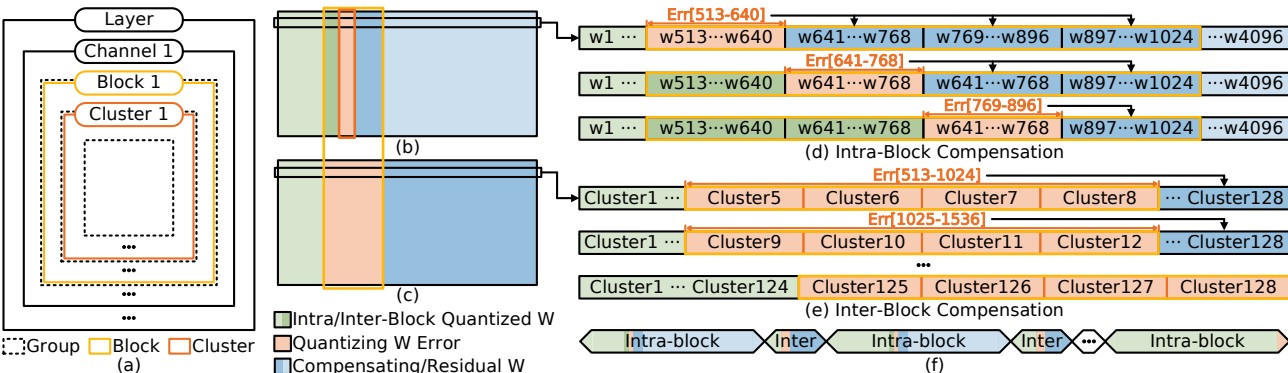

*Figure 3.* Structural hierarchy and error compensation mechanisms in STLA. The hierarchical organization of weight processing units ranging from Layer down to Cluster is illustrated in (a), providing the structural basis for the matrix state transitions visualized in (b) and (c). The framework manages quantization errors through a dual-scale strategy involving (d) localized intra-block compensation for fine-grained calibration and (e) long-range inter-block propagation for global error balancing, both of which are orchestrated according to the alternating sequential workflow presented in (f).

The STLA framework establishes the physical boundaries and computational units for weight quantization through five granular levels, as depicted in Figure 3(a). This hierarchy follows the scale: $Layer > Channel > \{Block, Group, Cluster\}$. A Layer represents the entire weight matrix $\mathbf{W}$, while a Channel refers to an independent row or column. Within each channel, Blocks are defined as coarse-grained segments for sequential processing to constrain the computational complexity of inverse Hessian updates. The Cluster serves as the atomic unit for spatial optimization, with its size constrained by $Cluster \leq Block$ to ensure structural alignment and eliminate zero-padding overhead. The Group defines the scope of shared quantization parameters (scale and zero-point); while its size only requires being smaller than a Channel, aligning it with Cluster size is preferred to suppress temporal distribution shifts during spatial alignment.

Under this hierarchical structure, the quantization process involves a dynamic evolution of matrix states. As visualized in the snapshots in Figure 3(b) and (c), the lookahead rounding errors (orange) generated by the stabilized quantized weights (green) are projected onto the remaining residual weights (blue) for absorption. This error management is executed across two distinct scales. On a local scale, intra-block compensation in Figure 3(d) leverages the high second-order correlations within the Hessian matrix to diffuse errors between adjacent segments (e.g., from $w_{513\sim640}$ to $w_{641\sim768}$), maximizing spatial error cancellation. Subsequently, inter-block compensation in Figure 3(e) addresses macro-scale error flow by propagating residual signals across block boundaries to distant clusters (e.g., toward Cluster 128), preventing error accumulation at channel extremities. The entire procedure is orchestrated through the alternating temporal workflow shown in Figure 3(f), which achieves high-fidelity quantization while maintaining hardware-friendly memory access patterns.

# C. Additional Experiments

We present further evaluations in this section to provide deeper insights into the performance and versatility of the STLA framework.

## C.1. Efficiency Comparison with the Decoupled Paradigm

Table 7 compares per-group STLA (group size = 256) with the decoupled method AESPA in a 3-bit per-channel quantization setting, as AESPA lacks support for group-wise PTQ. The results indicate that STLA achieves competitive PPL with significantly reduced runtime across all models. For OPT-1.3B, OPT-6.7B, and LLaMA2-7B, our method provides superior reconstruction fidelity while reducing GPU runtime by $1.7\times$, $4.6\times$, and $3.8\times$, respectively.

*Table 7.* PPL and Runtime comparison between STLA and AESPA under 3-bit weight-only quantization.

| METHOD | OPT-1.3B | | | OPT-6.7B | | | LLAMA2-7B | | |
|---|---|---|---|---|---|---|---|---|---|
| | WIKITEXT-2 | C4 | RUNTIME | WIKITEXT-2 | C4 | RUNTIME | WIKITEXT-2 | C4 | RUNTIME |
| FP16 | 14.62 | 16.07 | - | 10.86 | 12.71 | - | 5.47 | 7.26 | - |
| AESPA | 16.07 | 17.40 | 24.2 MIN | 11.35 | 13.42 | 3.4 HR | 6.45 | 8.51 | 2.6 HR |
| STLA | 15.24 | 17.12 | **13.7 MIN** | 11.06 | 13.17 | **43.9 MIN** | 6.09 | 7.99 | **40.6 MIN** |

## C.2. Comparison with Vector Quantization and QAT-Lite Methods

We further compare STLA with both scalar quantization and vector quantization methods on LLaMA2-7B under 2-bit weight quantization. As shown in Table 8, STLA achieves lower perplexity than PTQ baselines based on scalar quantization, including AWQ and QuIP, while also outperforming the PTQ version of QuIP# based on vector quantization. Compared with QAT-based vector quantization methods such as QuIP# and QTIP, our scalar PTQ method STLA achieves higher PPL but requires no additional training and fits within the memory of a single RTX 5090 GPU. This offers a practical trade-off between predictive accuracy and computational quantization cost.

*Table 8.* Performance (PPL) comparison between scalar quantization and vector quantization methods on LLaMA2-7B under 2-bit weight-only quantization.

| SETTING | METHOD | QUANTIZATION TYPE | WIKITEXT-2 | C4 |
|---|---|---|---|---|
| PTQ | AWQ | SCALAR QUANTIZATION | $2.29\times10^5$ | $1.73\times10^5$ |
| | QUIP | SCALAR QUANTIZATION | 21.30 | 24.06 |
| | QUIP# | VECTOR QUANTIZATION | 12.30 | 14.80 |
| | STLA | SCALAR QUANTIZATION | **10.02** | **12.31** |
| QAT | QUIP# | VECTOR QUANTIZATION | 6.66 | 8.35 |
| | QTIP | VECTOR QUANTIZATION | 6.47 | 8.12 |

## C.3. Per-channel 2-Bit PTQ Evaluation

*Table 9.* Quantization performance of per-channel 2-bit weight quantization on LLM models (Calibration data from C4).

| MODEL | METRIC | FP16 | GPTQ | AWQ | GPTAQ | QUIP | STLA |
|---|---|---|---|---|---|---|---|
| OPT-125M | WIKITEXT-2 | 27.66 | 758.91 | 3171.00 | 362.76 | 336.58 | **100.70** |
| | C4 | 25.89 | 389.06 | 2020.25 | 199.07 | 158.95 | **75.40** |
| OPT-6.7B | WIKITEXT-2 | 10.86 | 127.40 | 6573.67 | 78.98 | 22.33 | **18.57** |
| | C4 | 11.74 | 107.18 | 3226.28 | 64.98 | 21.62 | **20.01** |
| LLAMA2-7B | WIKITEXT-2 | 5.47 | 67.11 | $2.05\times10^5$ | 29.07 | 35.59 | **13.51** |
| | C4 | 7.26 | 116.86 | $1.62\times10^5$ | 21.56 | 33.91 | **14.91** |

To validate our 2-bit performance claims, it is valuable to enable a fair comparison with QuIP. Since QuIP does not natively support our focused group-wise PTQ setting, we establish a closest possible comparison by evaluating QuIP performance against channel-wise STLA. We evaluate STLA under per-channel 2-bit weight quantization alongside GPTQ, AWQ, and GPTAQ. To ensure a rigorous evaluation, both methods utilized the exact same C4 calibration set (128 segments, 2048-token length). As shown in Table 9, STLA consistently outperforms existing PTQ baselines across OPT-125M, OPT-6.7B, and LLaMA2-7B. These results demonstrate that STLA is not limited to group-wise quantization and remains highly effective even under coarser per-channel granularity.

## C.4. Large-Scale Model Evaluation

We further evaluate STLA on larger-scale LLMs, including LLaMA2-13B and LLaMA2-70B, under 2-bit group-wise weight quantization with group size 256. The calibration data are sampled from C4. These experiments are executed on a single NVIDIA H100 GPU (94GB). As shown in Table 10, STLA consistently outperforms existing group-wise PTQ baselines on both model scales, confirming its scalability. These results indicate that the proposed spatial-temporal lookahead alignment remains effective when scaling from 7B models to substantially larger LLMs.

*Table 10.* Performance comparison of 2-bit group-wise weight quantization on larger-scale LLMs. The group size is set to 256, and calibration data are sampled from C4.

| METHOD | LLaMA2-13B | | LLaMA2-70B | |
|---|---|---|---|---|
| | WIKITEXT-2 | C4 | WIKITEXT-2 | C4 |
| FP16 | 4.85 | 6.73 | 3.32 | 5.71 |
| GPTQ | 14.70 | 22.71 | 6.53 | 8.87 |
| AWQ | $1.25\times10^5$ | $9.84\times10^4$ | $7.25\times10^4$ | $6.52\times10^4$ |
| GPTAQ | 10.37 | 13.04 | 6.74 | 9.02 |
| STLA | **9.82** | **11.83** | **6.10** | **8.59** |

## C.5. Generative Benchmark Evaluation

In addition to perplexity and zero-shot classification accuracy, we further evaluate STLA on the few-shot generative GSM8K chain-of-thought reasoning benchmark. We use LLaMA2-7B under 3-bit group-wise weight quantization with group size 256, and the calibration data are sampled from C4. As shown in Table 11, STLA consistently outperforms existing group-wise PTQ baselines under both 8-shot and 4-shot settings, further strengthening the practical significance. Although low-bit quantization still causes a noticeable degradation compared with FP16, STLA preserves more reasoning capability than competing PTQ methods.

*Table 11.* Few-shot GSM8K-COT accuracy (↑) of 3-bit group-wise (group size = 256) weight quantization of STLA and existing group-wise PTQ methods on LLaMA2-7B. (Calibration data from C4)

| SHOT | FP16 | GPTQ | AWQ | GPTAQ | STLA |
|---|---|---|---|---|---|
| 8-SHOT | 13.04 | 7.28 | 7.81 | 8.42 | **9.63** |
| 4-SHOT | 7.73 | 4.55 | 4.55 | 4.70 | **4.85** |

## C.6. Hyperparameter and Calibration Robustness

To evaluate the reliability of STLA, we conducted experiments to assess its sensitivity to hyperparameter choices and robustness to calibration datasets. Unless otherwise specified, all results in this subsection are obtained under 3-bit weight quantization.

**Hyperparameter Sensitivity.** We conduct comprehensive experiments to assess the sensitivity of STLA to its key hyperparameters, including the learning rate (Table 12), the weight of the rounding loss (Table 13), and the number of optimization iterations (Table 14). The goal is to determine whether STLA requires precise hyperparameter tuning to achieve reliable quantization performance. Across multiple model families, we observe that STLA maintains consistent performance over a wide range of learning rates and rounding loss weights, while converging reliably after approximately 200 optimization iterations. These results suggest that STLA is inherently stable and robust, allowing users to use default

hyperparameter settings without sacrificing accuracy or convergence speed.

*Table 12.* Quantization performance of STLA under different learning rates for learning-based rounding. Calibration data are sampled from C4, the rounding loss weight is set to 1, and the number of iterations is set to 200.

| LR | OPT-125M | | OPT-1.3B | | OPT-6.7B | | LLAMA2-7B | | LLAMA3-8B | |
|---|---|---|---|---|---|---|---|---|---|---|
| | WIKITEXT-2 | C4 | WIKITEXT-2 | C4 | WIKITEXT-2 | C4 | WIKITEXT-2 | C4 | WIKITEXT-2 | C4 |
| 0.05 | 31.34 | 29.38 | 15.28 | 17.15 | 11.06 | 13.20 | 6.06 | 8.03 | 7.84 | 11.90 |
| 0.1 | 31.54 | 29.27 | 15.30 | 17.16 | 11.08 | 13.20 | 6.05 | 8.03 | 7.89 | 11.93 |
| 0.5 | 31.27 | 29.20 | 15.24 | 17.12 | 11.06 | 13.18 | 6.09 | 7.99 | 7.85 | 11.90 |
| 1.0 | 31.10 | 29.26 | 15.36 | 17.16 | 11.06 | 13.17 | 6.09 | 7.99 | 7.82 | 11.88 |

Table 12 shows the quantization performance of STLA across varying learning rates. The results indicate that STLA maintains consistent performance over a wide range of learning rates, with only minor fluctuations in perplexity across different model sizes and evaluation datasets.

*Table 13.* Quantization performance of STLA under different weights of the rounding loss for learning-based rounding. Calibration data are sampled from C4, and the number of iterations is set to 200.

| ROUNDING WEIGHT | OPT-1.3B, LR=0.5 | | OPT-6.7B, LR=1 | | LLAMA2-7B, LR=0.5 | | LLAMA3-8B, LR=1 | |
|---|---|---|---|---|---|---|---|---|
| | WIKITEXT-2 | C4 | WIKITEXT-2 | C4 | WIKITEXT-2 | C4 | WIKITEXT-2 | C4 |
| 0.5 | 15.45 | 17.16 | 11.04 | 13.18 | 6.12 | 7.99 | 8.11 | 11.93 |
| 1.0 | 15.24 | 17.12 | 11.06 | 13.17 | 6.09 | 7.99 | 7.82 | 11.88 |
| 1.5 | 15.38 | 17.14 | 11.11 | 13.17 | 6.13 | 7.99 | 7.86 | 11.92 |

Table 13 reports the effect of different rounding loss weights. STLA demonstrates stable behavior, with the performance remaining largely unchanged across a reasonable range of weights, suggesting that careful tuning of this parameter is not critical for achieving strong quantization results.

*Table 14.* Quantization performance and runtime of STLA under different numbers of optimization iterations for learning-based rounding. Calibration data are sampled from C4, and the rounding loss weight is set to 1.

| ITERS | OPT-1.3B, LR=0.5 | | | OPT-6.7B, LR=1 | | | LLAMA2-7B, LR=0.5 | | | LLAMA3-8B, LR=1 | | |
|---|---|---|---|---|---|---|---|---|---|---|---|---|
| | WKT-2 | C4 | RUNTIME | WKT-2 | C4 | RUNTIME | WKT-2 | C4 | RUNTIME | WKT-2 | C4 | RUNTIME |
| 100 | 15.32 | 17.19 | 10.5 MIN | 11.12 | 13.18 | 43.2 MIN | 6.11 | 8.00 | 41.0 MIN | 7.88 | 11.90 | 29.0 MIN |
| 200 | 15.24 | 17.12 | 13.2 MIN | 11.06 | 13.17 | 47.4 MIN | 6.09 | 7.99 | 45.5 MIN | 7.82 | 11.88 | 37.5 MIN |
| 300 | 15.43 | 17.11 | 16.0 MIN | 11.07 | 13.18 | 52.9 MIN | 6.07 | 8.04 | 49.4 MIN | 7.90 | 11.88 | 45.7 MIN |
| 400 | 15.38 | 17.12 | 18.6 MIN | 11.07 | 13.17 | 58.1 MIN | 6.08 | 7.99 | 53.2 MIN | 7.89 | 11.88 | 54.2 MIN |
| 500 | 15.40 | 17.10 | 20.8 MIN | 11.16 | 13.18 | 1.0 HR | 6.09 | 8.01 | 58.5 MIN | 7.94 | 11.88 | 1.0 HR |

Table 14 summarizes the impact of the number of optimization iterations. Increasing iterations beyond 200 provides marginal improvements in perplexity but increases runtime, indicating that 200 iterations offer a favorable trade-off between efficiency and accuracy.

Overall, these results demonstrate that STLA is robust to a wide range of hyperparameter choices. In particular, the learning rate and rounding loss weight do not require careful tuning, as varying them only leads to marginal changes in perplexity across different model families and evaluation datasets. For the optimization iteration, STLA achieves stable performance with 200 iterations, while further increasing the number of iterations yields little accuracy gain but incurs a clear runtime overhead. Therefore, we use 200 iterations as the default setting in experiments, which provides a practical balance between quantization quality and computational efficiency.

**Calibration Dataset Robustness.** We further evaluate the robustness of STLA to calibration data choices, including the calibration dataset and the calibration set size. Because both compensation-based and learning-based optimization use calibration examples, the calibration data distribution and sample size can directly affect the final quantized model. We

therefore conduct experiments with different calibration datasets in Table 15 and different numbers of calibration samples in Table 16. The results show that STLA is generally robust to these choices. Specifically, models tend to achieve the lowest perplexity on their corresponding calibration domains, and increasing the calibration size from 64 to 128 samples improves performance. However, further increasing the size to 256 samples mainly benefits the in-domain C4 evaluation while providing limited or even negative gains on WikiText-2, suggesting mild overfitting to the calibration distribution.

*Table 15.* Quantization performance of STLA under different calibration datasets ((128 segments, 2048-token length)).

| CALIBRATION DATASET | OPT-125M | | OPT-1.3B | | OPT-6.7B | | LLAMA2-7B | | LLAMA3-8B | |
|---|---|---|---|---|---|---|---|---|---|---|
| | WKT-2 | C4 | WKT-2 | C4 | WKT-2 | C4 | WKT-2 | C4 | WKT-2 | C4 |
| C4 | 31.01 | 29.26 | 15.24 | 17.12 | 11.06 | 13.17 | 6.09 | 7.99 | 7.82 | 11.88 |
| WIKITEXT-2 | 30.25 | 30.03 | 15.22 | 17.45 | 11.01 | 13.37 | 5.83 | 8.29 | 7.22 | 12.62 |

Table 15 compares quantization performance when using different calibration datasets. Models achieve the best performance on their respective calibration domains, while cross-dataset calibration leads to slight increases in perplexity, highlighting the importance of selecting representative calibration data.

*Table 16.* Quantization performance and runtime of STLA under different calibration dataset sizes. Calibration data are sampled from C4.

| SIZE | OPT-1.3B | | | OPT-6.7B | | | LLAMA2-7B | | | LLAMA3-8B | | |
|---|---|---|---|---|---|---|---|---|---|---|---|---|
| | WKT-2 | C4 | RUNTIME | WKT-2 | C4 | RUNTIME | WKT-2 | C4 | RUNTIME | WKT-2 | C4 | RUNTIME |
| 64 | 15.36 | 17.22 | 11.8 MIN | 11.19 | 13.20 | 34.8 MIN | 6.10 | 8.03 | 32.3 MIN | 8.12 | 12.04 | 30.8 MIN |
| 128 | 15.24 | 17.12 | 13.2 MIN | 11.06 | 13.17 | 47.4 MIN | 6.09 | 7.99 | 45.5 MIN | 7.82 | 11.88 | 37.5 MIN |
| 256 | 15.36 | 17.07 | 15.9 MIN | 11.13 | 13.14 | 58.2 MIN | 6.08 | 7.98 | 56.4 MIN | 7.96 | 11.76 | 44.9 MIN |

Table 16 reports the effect of varying the number of calibration samples. Increasing the calibration set from 64 to 128 samples improves performance, while further increasing it to 256 mainly reduces in-domain (C4) perplexity with limited gains on the out-of-domain dataset (WikiText-2) and incurs additional computational cost. This indicates that 128 samples provide a good balance between accuracy and runtime.

### C.7. Evaluation of Temporal Conflicts in Quantization Parameters

Table 17 highlights the critical role of maintaining consistency between group and cluster sizes. Learning-based rounding is disabled in this experiment to isolate and demonstrate the adverse effects of temporal conflicts in quantization parameters caused by vector-by-vector compensation. Using element-wise compensation with a cluster size of 1 leads to significant shifts in the weight distribution. These shifts invalidate the initial quantization parameters and result in degraded perplexity. In contrast, matching the cluster size to the group size ensures that decisions remain aligned with the local distribution throughout the process. Consequently, STLA can be reliably deployed across various hardware constraints, offering a flexible yet principled approach to model compression without the risk of distribution-induced performance degradation.

*Table 17.* Analysis of weight distribution shift via synchronized granularity with learning-based rounding disabled.

| GROUP | CLUSTER | OPT-125M | | LLAMA2-7B | |
|---|---|---|---|---|---|
| | | WIKITEXT-2 | C4 | WIKITEXT-2 | C4 |
| 128 | 1 | 34.37 | 31.19 | 6.24 | 8.20 |
| 128 | 128 | 34.12 | 30.35 | 6.18 | 8.24 |
| 256 | 1 | 35.15 | 31.52 | 6.33 | 8.33 |
| 256 | 256 | 33.04 | 30.89 | 6.30 | 8.34 |

### C.8. Evaluation of Different Cluster Size

Table 18 reveals that STLA maintains a highly stable perplexity profile even as the quantization scale shifts. Specifically, by matching the cluster size to the respective group size, the framework ensures that the compensatory adjustments are

strictly localized, regardless of whether a fine-grained ($G = 128$) or a coarser ($G = 256$) strategy is employed. This granularity-agnostic effectiveness is particularly evident in the LLaMA-2-7B results. Increasing the configuration to 256 achieves a significant reduction in runtime (from 57.2 to 40.6 minutes) with negligible impact on accuracy. Such flexibility allows STLA to serve as a versatile solution, capable of being optimized for either maximum inference speed or peak model fidelity depending on the target deployment environment.

*Table 18.* Evaluation of STLA across different group and cluster size configurations under 3-bit weight-only quantization.

| Group | Cluster | OPT-125M | | | LLaMA2-7B | | |
|---|---|---|---|---|---|---|---|
| | | Wikitext-2 | C4 | Runtime | Wikitext-2 | C4 | Runtime |
| 128 | 128 | 31.37 | 28.97 | 2.4 min | 6.01 | 7.95 | 57.2 min |
| 256 | 256 | 31.01 | 29.26 | 1.5 min | 6.09 | 7.99 | 40.6 min |

