# OpenReview forum: "STLA: Spatiotemporal Lookahead Alignment for Post-Training Quantization"
_ICML.cc/2026/Conference — ICML 2026 regular_

### Official Review · Reviewer_2Hmm · 2026-03-09

**Soundness:** 3
**Presentation:** 3
**Significance:** 3
**Originality:** 3
**Overall Recommendation:** 4
**Confidence:** 4

**Summary:**

This work aims to improving post-training quantization for llms by resolving the conflicts bwtween learning-based adaptive rounding and compensation-based second-order solvers. Authors identify two major chanllenges: temporal misalignment (conflicts between rounding assumptions and subsequent compensation) and spatial misalignment (local optimization objectives that ignore global curvature structure). The core technical contributions include: (1) a lookahead global objective derived via the Schur complement of the inverse Hessian to align local cluster-wise rounding with global reconstruction error; (2) a coupled cluster-wise scheme that collocates learning-based rounding and compensation to avoid RTN-related temporal conflicts; and (3) a Hessian-guided SpinCluster strategy to group strongly correlated weights for improved intra-cluster error cancellation. Experiments on OPT and LLaMA families in 3-bit and 2-bit weight-only PTQ demonstrate consistent perplexity and zero-shot accuracy improvements over GPTQ, AWQ, and GPTAQ, particularly in the low-bit regime.

**Compliance With Llm Reviewing Policy:**

Affirmed.

**Key Questions For Authors:**

None

**Limitations:**

Yes

**Strengths And Weaknesses:**

Strengths
1.	The manuscript is well-structured, with motivations, problem formulation, methodology, and experimental evaluation presented in a logical and coherent manner.
2.	The results are compelling. STLA demonstrates consistent improvements over GPTQ/AWQ/GPTAQ- baselines across both perplexity and zero-shot evaluation benchmarks.

Weaknesses
1.	GPTQ-style methods assume the Hessian (computed from full-precision weights) remains constant throughout the quantization process. However, STLA's coupled optimization might amplify errors since its learning and compensation interact more tightly and lacks verification.
2.	The SpinCluster algo uses a greedy heuristic. Further analysis or theoretical bound are suggested.
3.	Although STLA outperforms GPTQ family baselines, authors did not compare STLA against recent hybrid or more advanced PTQ/QAT-lite methods beyond these families. It would strengthen the claim of state-of-the-art performance to include broader comparisons.

---

> ### Author Rebuttal · Authors · 2026-03-31
>
> We thank the reviewer for recognizing STLA's coherent methodology and compelling results. Based on your suggestions, we have: (1) clarified Hessian stability in STLA; (2) added an analysis of SpinCluster’s theoretical bound; and (3) expanded experimental comparisons to reinforce the claim of state-of-the-art performance. **New results (Tables R1, R10) can be viewed at https://anonymous.4open.science/r/STLA_Rebuttal.** A detailed response to each point follows.
>
> **Weakness (W) and Question (Q)**
>
> > W1: Concern of Error Amplification under Constant Hessian
>
> **A1**: We thank the reviewer for raising this insightful question. We would like to **clarify STLA’s efforts on mitigate the Hessian error throughout the quantization process on two distinct levels: intra-layer and cross-layer.** Within a layer, the Hessian itself remains unchanged during compensation or learning. The main source of Hessian staleness arises from cross-layer error propagation, as quantizing an earlier layer alters the input to subsequent layers.
> - **Intra-Layer**: Within a single layer, the main challenge is residual mismatch. Compensation is computed for one residual, while subsequent discrete rounding may produce another. STLA mitigates this mismatch by coupling the current rounding residual with its future compensation under a shared global objective, supported by rigorous theoretical analysis in Sections 4.4/4.5 and Appendix A.1.3/A.2. **Its spatiotemporally coupled objective directly minimizes global reconstruction error, thereby reducing error propagation to later layers.**
> - **Cross-Layer**: To prevent error amplification across the network, STLA does not assume a static, full-precision Hessian globally. Instead, it computes the Hessian for each layer at runtime using the newly quantized outputs from the previous layer. To further reduce cross-layer error propagation, STLA incorporates an input-side correction proposed by GPTAQ [1]. By **combining dynamic runtime Hessians with this input-side correction, STLA effectively mitigates error amplification.**
>
> Overall, we emphasize that **STLA’s tightly coupled design mitigates rather than amplifies error**.
>
> [1] Li, Y., Yin, R., Lee, D., Xiao, S., and Panda, P. GPTAQ: Efficient Finetuning-Free Quantization for Asymmetric Calibration. In Proceedings of the 42nd International Conference on Machine Learning, volume 267, pp. 36690–36706, 13–19 Jul 2025.
>
> > W2: SpinCluster Theoretical Bound Analysis
>
> **A2**: Thank you for the valuable suggestion. We have **added an analysis of SpinCluster’s theoretical bound** and would like to clarify that, while SpinCluster is greedy, it is not arbitrary. Its design is directly motivated by the second-order objective. Let
>
> $L_{full}(\Delta w)=\sum_i H_{ii}\Delta w_i^2+\sum_{i\neq j} H_{ij}\Delta w_i\Delta w_j$
>
> be the full quadratic objective, and let $L_{cluster}$ be the cluster-restricted version that keeps only within-cluster interactions. If $E_{cut}$ denotes the set of cross-cluster pairs, then the gap between the two objectives is exactly the omitted cross-cluster Hessian interaction:
>
> $L_{full}(\Delta w)-L_{cluster}(\Delta w)=\sum_{(i,j)\in E_{cut}} H_{ij}\Delta w_i\Delta w_j$.
>
> Hence, we obtain the following theoretical bound:
>
> $|L_{full}(\Delta w)-L_{cluster}(\Delta w)| \le \sum_{(i,j)\in E_{cut}} |H_{ij}|\,|\Delta w_i|\,|\Delta w_j| \le \epsilon^2 \sum_{(i,j)\in E_{cut}} |H_{ij}|$,
>
> assuming $|\Delta w_i|\le\epsilon$. **This bound explicitly justifies SpinCluster, as grouping strongly interacting weights into the same cluster reduces omitted cross-cluster terms and better preserves the full objective.**
>
> > W3: Broader Comparison with Other Methods
>
> **A3**: We agree that broader comparisons would strengthen the SOTA claim and have expanded our empirical evaluation accordingly. Beyond the original GPTQ-family baselines, we now **include more advanced low-bit quantization methods**, including **PTQ methods** QuIP, QuIP# (PTQ version) and AWQ, as well as **QAT-lite methods** QuIP# and QTIP. As shown in **Table R10**, STLA outperforms PTQ baselines, while QAT-lite methods achieve stronger results. We believe this is a fair outcome: QAT methods require additional training, access to the original data and substantial high-performance GPU resources, making them more expensive and less accessible in practice.  In contrast, **STLA is a pure PTQ method that requires no retraining, remains fast and resource-efficient, and achieves state-of-the-art performance within the PTQ setting, demonstrating clear practical value.**
>
> We also evaluate STLA in combination with **advanced rotation-based PTQ methods**, including QuaRot and SpinQuant, which provide further gains. **Table R1** shows that STLA is compatible with rotation-based methods and achieves superior accuracy, confirming its efficacy in realistic deployment.
>
> Overall, these new experiments position STLA more clearly with respect to recent advanced PTQ methods and further highlight its practical benefits.

---

> > ### Author Rebuttal · Reviewer_2Hmm · 2026-04-02
> >
> > Fully resolved.

---

> > > ### Author Response · Authors · 2026-04-08
> > >
> > > We are glad to hear that all of the concerns were addressed. We appreciate your time and effort for the review.

---

### Official Review · Reviewer_RPF2 · 2026-03-11

**Soundness:** 3
**Presentation:** 3
**Significance:** 3
**Originality:** 3
**Overall Recommendation:** 5
**Confidence:** 3

**Summary:**

This paper studies low-bit weight-only post-training quantization (PTQ) for large language models. The main motivation is that existing decoupled pipelines, which first perform error compensation and then apply learnable rounding, suffer from what the authors call temporal and spatial misalignment. To address this, the paper proposes STLA, a unified PTQ framework built around three components: a lookahead global objective derived via a Schur complement argument, a cluster-wise coupling of learnable rounding and compensation, and a Hessian-based clustering method (SpinCluster) that groups strongly correlated weights. Experiments on OPT and LLaMA-family models show improvements over GPTQ, AWQ, and GPTAQ in 3-bit and especially 2-bit group-wise weight PTQ.

**Compliance With Llm Reviewing Policy:**

Affirmed.

**Final Justification:**

The paper is technically sound and now substantially better supported than in the initial submission. On soundness, the core formulation (lookahead objective with coupled compensation/rounding and clustering) was already coherent, and the rebuttal strengthened empirical support by adding controlled 2-bit comparisons that include QuIP under matched calibration conditions. On significance, the added 13B and 70B results materially improve confidence that the method scales beyond small-to-mid-size models, which was my main practical concern. On originality, I still view this as moderate rather than radical novelty, but the integration of temporal/spatial alignment and the associated optimization design is meaningful and likely useful for follow-up work. On clarity, the authors acknowledged over-strong phrasing and committed to more precise wording, which improves trustworthiness.

**Key Questions For Authors:**

1. Can the authors add a comparison against QuIP for the 2-bit setting? If a direct comparison is not possible because QuIP does not match the exact group-wise PTQ setting used here, the paper should clearly explain that and provide the closest possible comparison or an additional discussion that situates STLA relative to QuIP.

2. Can the authors provide results on at least one larger model, ideally at the 13B scale or beyond (and, if feasible, even 70B), or otherwise clarify the scalability of the method beyond the current model sizes? Since the motivation is practical low-bit PTQ for LLM deployment, additional evidence on a larger model would strengthen the significance of the work.

**Limitations:**

yes

**Strengths And Weaknesses:**

### Strengths

A key strength of the paper is that the problem formulation is clear and well motivated. The paper does not simply present another PTQ variant, but instead identifies a concrete failure mode in existing decoupled pipelines and organizes it into two forms of misalignment: temporal and spatial. This framing gives the method a coherent narrative and makes the proposed components feel principled rather than ad hoc.

The technical core is also reasonably strong. The lookahead global objective is a meaningful reformulation of the local quantization problem under the assumption that future compensation will be applied optimally, and the resulting derivation appears conceptually sound. The use of cluster-wise coupling and the Cholesky-based implementation also makes the method look practically thoughtful, rather than purely theoretical.

Empirically, the paper does a decent job of validating its main ingredients. The ablations test the effect of coupling, the global objective, and SpinCluster, and the main results show consistent gains over GPTQ, AWQ, and GPTAQ. In particular, the 2-bit results are strong relative to the baselines included in the paper, which is important because this low-bit regime is where robustness matters most.

### Weaknesses

My main concern is that the paper strongly emphasizes its 2-bit performance, but **does not compare against QuIP**, which is hard to ignore in this regime. The paper itself cites QuIP as related work, and QuIP is one of the most relevant references when making strong claims about very low-bit LLM quantization. As a result, the current experimental picture feels incomplete. Even if a direct apples-to-apples comparison is difficult because of different quantization settings, the paper should either include such a comparison or explain clearly why it is not possible. Without that, the paper’s low-bit PTQ claims feel less convincing than they could be.

A second weakness is the limited model scale of the evaluation. The experiments cover OPT-125M/1.3B/6.7B, LLaMA2-7B, and LLaMA3-8B, which is useful, but still somewhat modest for a paper motivated by practical LLM deployment. I would be more convinced about the broader significance of the method if at least one larger model, such as a **13B-scale or even 70B-scale model**, were included. The zero-shot evaluation is also fairly limited.

Finally, some of the empirical claims in the paper feel slightly stronger than what the tables fully support. The overall trend is favorable to the proposed method, but in a few cases the advantage of the global objective is not as uniform as the presentation suggests. This is not a fatal issue, but the writing would benefit from being a bit more careful in how broadly it states these conclusions.

---

> ### Author Rebuttal · Authors · 2026-03-31
>
> Thank you for your thoughtful and constructive comments. We appreciate your positive feedback on our methodological contributions, technical core, and robustness in the low-bit regime. In response, we have added new **2-bit comparisons against QuIP**, expanded evaluations to **larger-scale 13B and 70B models**, and refined the writing tone.
>
> **Weakness (W) and Question (Q)**
>
> > W1&Q1: 2-bit Fair Comparison with QuIP
>
> **A1**: We thank the reviewer for this constructive feedback. We agree that a comparison with QuIP is valuable for validating our 2-bit claims. Because QuIP does not natively support our focused group-wise PTQ setting, **we established a closest possible comparison by comparing QuIP performance against channel-wise STLA**. To ensure a rigorous evaluation, both methods utilized the exact same C4 calibration set (128 segments, 2048-token length). The results in Table R8 demonstrate that STLA achieves better accuracy than QuIP under these matched conditions, further strengthening our low-bit PTQ claims.
>
> *Table R8*: Quantization Performance (PPL) of **per-channel** 2-bit weight quantization on LLM models. (Calibration data from C4)
>
> | MODEL | METRIC | FP16 | GPTQ | AWQ | GPTAQ | QUIP | **STLA** |
> | :--- | :--- | :---: | :---: | :---: | :---: | :---: | :---: |
> | **OPT-125M** | WKT-2(↓) | 27.66 | 758.91 | 3171.00 | 362.76 | 336.58 | **100.70** |
> | | C4(↓) | 25.89 | 389.06 | 2020.25 | 199.07 | 158.95 | **75.40** |
> | **OPT-6.7B** | WKT-2(↓) | 10.86 | 127.40 | 6573.67 | 78.98 | 22.33 | **18.57** |
> | | C4(↓) | 11.74 | 107.18 | 3226.28 | 64.98 | 21.62 | **20.01** |
> | **LLAMA2-7B** | WKT-2(↓) | 5.47 | 67.11 | 2.05e5 | 29.07 | 35.59 | **13.51** |
> | | C4(↓) | 7.26 | 116.86 | 1.62e5 | 21.56 | 33.91 | **14.91** |
>
> > W2&Q2: Lack of Evaluation on Larger-Scale Models (13B/70B)
>
> **A2**: We thank the reviewer for the insightful feedback. We have **added new experiments on larger models (13B and 70B)** using a single NVIDIA H100 GPU (94GB). The results in Table R9 show that STLA remains effective at substantially larger scales and achieves state-of-the-art performance, confirming its scalability.
>
> *Table R9*: Performance (PPL) of 2-bit group-wise (group size = 256) weight quantization of STLA and existing group-wise PTQ methods on **larger-scale LLM models**. (Calibration data from C4)
> | | LLAMA2-13B | LLAMA2-70B |
> | :--- | :---: | :---: |
> | METHOD | WKT-2(↓) &emsp; C4(↓) | WKT-2(↓) &emsp; C4(↓) |
> | FP16  | 4.85 &emsp; 6.73 | 3.32 &emsp; 5.71 |
> | GPTQ  | 14.70 &emsp; 22.71 | 6.53 &emsp; 8.87 |
> | AWQ   | 1.25e5 &emsp; 9.84e4 | 7.25e4 &emsp; 6.52e4 |
> | GPTAQ | 10.37 &emsp; 13.04 | 6.74 &emsp; 9.02 |
> | **STLA**  | **9.82** &emsp; **11.83** | **6.10** &emsp; **8.59** |
>
> Additionally, we expanded our evaluation to **include the few-shot generative GSM8K-CoT benchmark** in Table R7. Because zero-shot results are too weak to be informative here, we report 4-shot and 8-shot performance. STLA achieves the best accuracy across these settings, further strengthening the practical significance.
>
> *Table R7*: Few-shot **GSM8K-COT** accuracy of 3-bit group-wise (group size = 256) weight quantization of STLA and existing group-wise PTQ methods on LLaMA2-7B. (Calibration data from C4)
>
> | METHOD | 8-shot(↑) | 4-shot(↑) |
> | :--- | :---: | :---: |
> | FP16  | 13.04 | 7.73 |
> | GPTQ  | 7.28  | 4.55 |
> | AWQ   | 7.81  | 4.55 |
> | GPTAQ | 8.42  | 4.70 |
> | **STLA** | **9.63** | **4.85** |
>
> > W3: Writing Tone Refinement
>
> **A3**: We thank the reviewer for the valuable suggestion. We agree that while the overall empirical trend supports the proposed components of STLA, the individual benefits are not strictly uniform across all configurations, they achieve their peak performance when combined synergistically. We will revise the wording to replace absolute claims with more precise language, improving clarity and rigor.

---

> > ### Author Rebuttal · Reviewer_RPF2 · 2026-04-04
> >
> > Based on the authors' rebuttal, I consider my primary concerns to be adequately addressed. In particular, (1) for the 2-bit setting, the authors added a QuIP comparison under matched calibration conditions, which materially strengthens the low-bit claim; (2) regarding scalability, they added experiments on 13B and 70B models, directly addressing my concern about limited evaluation scale; and (3) they acknowledged that improvements are not uniform in all configurations and committed to revising overly strong wording.
> >
> > Overall, I regard the core concerns from my original review as sufficiently resolved.

---

> > > ### Author Response · Authors · 2026-04-08
> > >
> > > We thank the reviewer for the constructive suggestions and the positive assessment of our work. We are pleased to have addressed the core concerns regarding the low-bit claims and scalability, and we will incorporate these improvements and the refined wording about the synergistic effect into the revision.

---

### Official Review · Reviewer_phes · 2026-03-12

**Soundness:** 3
**Presentation:** 2
**Significance:** 3
**Originality:** 2
**Overall Recommendation:** 4
**Confidence:** 4

**Summary:**

This paper investigates the optimization inconsistency that arises when combining two widely used PTQ paradigms: learning-based rounding and compensation-based quantization. The authors attribute this issue to temporal misalignment, where later optimization steps invalidate the assumptions made in earlier stages (e.g., fixed RTN-based quantization versus learnable rounding), and spatial misalignment, where local rounding decisions fail to capture the global structure of the loss landscape. To address these issues, the paper proposes STLA, a unified PTQ framework that replaces local reconstruction objectives with a lookahead global objective that accounts for future error compensation. The method further performs cluster-wise joint optimization of rounding and compensation guided by Hessian-based clustering. Experiments on several LLM benchmarks show that STLA improves quantization accuracy over existing PTQ baselines under comparable bit-width settings.

**Compliance With Llm Reviewing Policy:**

Affirmed.

**Final Justification:**

The paper addresses an important and practical problem in PTQ by identifying optimization inconsistencies when combining learning-based rounding and compensation-based methods, and proposing a unified framework to resolve them. I found the perspective on spatiotemporal misalignment to be insightful, and the overall approach technically sound.

The rebuttal further strengthens the paper by addressing several of my main concerns. In particular, the additional experiments demonstrating compatibility with rotation-based methods (e.g., QuaRot, SpinQuant) significantly improve the practical relevance of the work. The added robustness analysis on hyperparameters and calibration data also makes the method more credible for real-world deployment.

Overall, I have a positive view of this paper, and the rebuttal further reinforces its strengths.

**Key Questions For Authors:**

- The paper initially focuses on the compensate-then-learn formulation. What happens if the order is reversed (i.e., learn-then-compensate)? In that case, the compensation step would be applied after the learned rounding decisions, which may partially mitigate the temporal misalignment described in the paper.
- In Figure 1(c), how would other variants perform, such as group-wise learn rounding or group-wise compensate-then-learn?
- How does STLA compare with vector-quantization-based approaches or rotation-based quantization methods in terms of the accuracy–efficiency trade-off?

**Limitations:**

yes

**Strengths And Weaknesses:**

**Strengths**

- Jointly considering compensation-based approaches (e.g., GPTQ) and learning-based rounding approaches (e.g., AdaRound) is an interesting direction. The paper provides an insightful interpretation of the challenges that arise when combining these paradigms through the lens of spatiotemporal misalignment.
- The paper provides a comprehensive background discussion, covering not only LLM PTQ methods but also earlier work from CNN quantization such as AdaRound and BRECQ. This helps establish the motivation and context of the work clearly.
- The ablation studies (Table 1, Appendix C.2) is informative and helps clarify the contribution of STLA framework.
- Calibration time is an important practical consideration for PTQ methods. The paper provides both complexity analysis and empirical calibration time measurements, which strengthens the practical relevance of the work.

**Weaknesses**

- From a methodological perspective, the proposed approach appears to build heavily on existing ideas from AdaRound, OBQ, and GPTQ. The modifications to the objective and optimization procedure may therefore appear incremental. In addition, the proposed SpinCluster strategy is conceptually related to the Hessian-based ordering heuristic used in GPTQ implementations (e.g., the --act-order option). To better establish novelty, it would be helpful to compare against stronger baselines formed by naïvely combining existing methods. For example, configurations such as channel-wise STLA or group-wise compensate-then-learn are not included in Figure 1(c) or Table 1.
- The GPTAQ paper reports that combining quantization with rotation-based techniques such as QuaRot and SpinQuant achieves lower perplexity than both the results reported in Table 2 of this paper and those obtained using STLA alone. Rotation-based approaches are widely adopted in modern quantization frameworks (e.g., NVIDIA ModelOPT). However, the paper does not evaluate STLA in combination with such techniques, even though they appear to be compatible. Including such results would help clarify the practical performance of STLA in realistic deployment settings.
- Since STLA introduces learnable parameters for rounding optimization, the method may be sensitive to hyperparameters and calibration data selection. However, the paper does not discuss hyperparameter sensitivity or robustness with respect to the calibration dataset.
- While the paper focuses on uniform quantization, recent work based on vector quantization (e.g., QuIP#, QTIP) has demonstrated stronger performance in several settings. A clearer discussion of why uniform quantization remains preferable in practice would strengthen the paper.
- The evaluation uses somewhat dated models and tasks, and primarily focuses on prefill-oriented metrics (e.g., next-token prediction). It would be helpful to evaluate the method on more recent models and generative benchmarks (e.g., GSM8K), where the practical impact of improved quantization accuracy could be better demonstrated.

---

> ### Author Rebuttal · Authors · 2026-03-31
>
> Thank you for your valuable feedback. We have clarified our methodological contributions and incorporated suggested experiments. **New results (Fig. R1, Tables R1-R7) can be viewed at https://anonymous.4open.science/r/STLA_Rebuttal.** Detailed responses are provided as:
>
> **Weakness (W) and Question (Q)**
>
> > W1.1: STLA vs. Existing Methods
>
> **A1**: We appreciate the opportunity to clarify our methodological contributions. Building upon important ideas from AdaRound and GPTQ, we respectfully argue that STLA is **NOT** limited to modifying the optimization objective and procedure, but instead **the first to identify and resolve a fundamental failure in decoupled PTQ pipelines**. Core contributions are:
> 1. identify the root cause of failure in decoupled PTQ pipeline as temporal and spatial misalignments;
> 2. address temporal misalignments by collocating learning and compensation phases;
> 3. propose a novel lookahead global objective that anticipates future compensation, overcoming local subspace optima caused by spatial misalignment;
> 4. introduce SpinCluster, the first clustering strategy to pre-structure for maximizing intra-cluster error cancellation under the coupled framework.
>
> Therefore, novelty lies in **how STLA fundamentally redefines the interaction between learning and compensation in a synergistic manner**.
>
> > W1.2: SpinCluster vs. act-order
>
> **A2**: We clarify that SpinCluster is fundamentally different from act-order. While act-order relies only on diagonal $H_{ii}$ to prioritize sensitive columns, SpinCluster is the **first to leverage off-diagonal Hessian $H_{ij}$ to group strongly correlated weights for maximizing intra-cluster error cancellation**. This makes SpinCluster conceptually novel, designed to support the collocated learning and compensation.
>
> > W1.3&Q2: Comparison with Other Combinations
>
> **A3**: We have added **group-wise learn rounding, group-wise compensate-then-learn, and channel-wise STLA**, updating Fig. R1. The results show that STLA consistently outperforms these variants, **translating its theoretical novelty into practical improvements in both accuracy and efficiency**.
>
> > W2&Q3: Compatibility with Rotation
>
> **A4**: **STLA is fully compatible with rotation-based methods** including QuaRot and SpinQuant. In Table R1, new W3A4 and W2A4 comparison experiments show that combining STLA with rotation achieves superior accuracy, confirming efficacy in realistic deployment.
>
> > W3: Hyperparameter and Calibration Robustness
>
> **A5**: We have **added robustness analysis experiments**:
> 1. Hyperparameter: Experiments on different learning rate (Table R2), weight of rounding loss (Table R3), and number of iterations (Table R4). STLA exhibits stability across varied hyperparameters, converging reliably at around 200 iterations.
> 2. Calibration data: Experiments on different calibration dataset (Table R5) and size (Table R6). Models achieve the lowest PPL on their respective calibration domains. Increasing the calibration size from 64 to 128 improves performance, while exceeding 128 samples (e.g., from C4) leads to overfit, reducing PPL on C4 but degrading accuracy on WikiText-2.
>
> > W4&Q3: Comparison with Vector Quantization
>
> **A6**: While vector quantization (VQ) such as QuIP# and QTIP offers higher accuracy than uniform quantization (UQ) according to Shannon’s Rate-Distortion theory, UQ remains preferable in practice due to the trade-off:
> - **Production Complexity**: VQ requires constructing, storing, and searching large codebooks, resulting in a computationally expensive and resource-intensive training process, whereas UQ methods like STLA utilize highly efficient scalar process, with simple and fast production.
> - **Hardware Friendliness**: VQ deployment is heavily bottlenecked in memory and latency due to codebook LUTs and the lack of ALU support. In contrast, UQ is fully parallelizable (SIMD) and maps directly onto modern hardware (e.g., INT4 Tensor Cores), offering higher throughput and lower power. Overall, UQ is supported by most AI accelerators, whereas VQ is rarely supported in hardware and requires costly software-level efforts.
>
> We will include this trade-off in revised manuscript to strengthen the practical benefit of STLA.
>
> > W5: Evaluation on Generative Benchmarks
>
> **A7**: We have added experiments on GSM8K generative benchmark in Table R7, showing practical impact of the improved accuracy achieved by STLA.
>
> > Q1: Learn-Then-Compensate Analysis
>
> **A8**: We respectfully address a critical misunderstanding. **Learn-then-compensate does NOT mitigate temporal misalignment, it exacerbates it**. Element-wise rounding decisions learned on initial weights are entirely invalidated by subsequent compensation updates. As compensate-then-learn struggles to match learn-only, learn-then-compensate performs even worse than compensate-only. **The problem is not which step comes first, but the decoupled nature of the pipeline itself, which highlights STLA's core motivation and contribution.**

---

> > ### Author Rebuttal · Reviewer_phes · 2026-04-01
> >
> > Thank you for the rebuttal. Several of my main concerns regarding practicality, robustness, and empirical validation have been adequately addressed.
> >
> > First, the newly added experiments demonstrating that STLA is compatible with rotation-based methods (e.g., QuaRot, SpinQuant) and further improves their performance significantly strengthen the paper. This is an important result, as rotation-based techniques are widely used in modern quantization pipelines. Demonstrating that STLA can be combined with such approaches to achieve higher accuracy is essential and further highlights the practical relevance of the proposed method.
> >
> > Second, the additional robustness analysis on hyperparameters and calibration data alleviates concerns about the stability of the approach. The results indicate that STLA is not overly sensitive to learning rate, loss weighting, or iteration count, and that its behavior under varying calibration datasets and sizes is well-characterized. This makes the method more credible for real-world deployment, where such sensitivities are often a major bottleneck.
> >
> > Regarding vector quantization, I generally agree with the authors’ discussion on its practical limitations. However, recent methods such as QuIP# have demonstrated tangible speedups in practice. Therefore, simply arguing that vector quantization is “complex” may not be fully convincing to all readers. I encourage the authors to incorporate the discussion provided in the rebuttal into the main paper, with a more nuanced treatment of this trade-off.
> >
> > Based on these improvements, I am updating my overall score from 3 to 4.

---

> > > ### Author Response · Authors · 2026-04-08
> > >
> > > Thank you for the positive feedback and the insightful suggestion.
> > >
> > > In the revised manuscript, we will include a nuanced comparison with vector quantization, highlighting the respective trade-offs.

---

### Decision · Program_Chairs · 2026-04-30

**Decision:**

Accept (regular)

**Comment:**

This paper introduces STLA, a post-training quantization framework that uses a unified spatiotemporal optimization scheme to align learning-based rounding with compensation-based quantization. Its core contributions include a lookahead global objective, coupled cluster-wise optimization, and a Hessian-guided clustering strategy designed to enhance low-bit performance.

The submission received consistent support from all three reviewers, earning final scores of 4, 5, and 4 (average 4.33). Reviewers highlighted the well-motivated problem statement, technical soundness, and strong empirical results. While initial concerns were raised regarding novelty and the need for larger-scale evaluations, the rebuttal effectively addressed these by providing comparisons with stronger baselines like QuIP, testing on 70B models, and demonstrating compatibility with rotation-based methods.

With all major concerns resolved and the authors providing convincing experimental evidence, the paper offers a solid and practically relevant contribution to the field. We recommend Accept.